



# Review article: 30 years of airborne radar surveys on the Antarctic and Greenland ice sheets by the Alfred Wegener Institute

Steven Franke[1,2,*], Daniel Steinhage[2,*], Veit Helm[2,*], Tobias Binder[2,†], Uwe Nixdorf[2], Heinrich Miller[2], Angelika Humbert[2,3], Daniela Jansen[2], Graeme Eagles[2], Hannes Eisermann[2], Wilfried Jokat[2], Antonia Ruppel[3], Reinhard Drews[1], Alexandra Zuhr[1], Amelie Driemel[2], Andreas Walter[2], Peter Konopatzky[2], Robin Heß[2,‡], Antonie Haas[2], Roland Koppe[2], Pascal H. Andreas[2], and Olaf Eisen[2,4]

[1]Department of Geosciences, Tübingen University, Tübingen, Germany
[2]Alfred Wegener Institute, Helmholtz Centre for Polar and Marine Research, Bremerhaven, Germany
[3]Federal Institute for Geosciences and Natural Resources (BGR), Hanover, Germany
[4]Faculty of Geosciences, University of Bremen, Bremen, Germany
[†]Now at: Atlas Elektronik GmbH, Hamburg, Germany
[‡]Formerly at: Alfred Wegener Institute, Helmholtz Centre for Polar and Marine Research, Bremerhaven, Germany
[*]These authors contributed equally to this work

**Correspondence:** Steven Franke (steven.franke@uni-tuebingen.de) and Olaf Eisen (olaf.eisen@awi.de)

**Abstract.** The Alfred Wegener Institute, Helmholtz Centre for Polar and Marine Research (AWI), has conducted airborne radar campaigns since 1994 across Antarctica and Greenland, utilizing six different radar systems to study ice sheets and their interactions with climate, ocean and the solid Earth. In this review article, we describe AWI's airborne radar systems and their deployments over the Greenland and Antarctic Ice Sheet. Moreover, we summarize application and usage of AWI radar systems, which provided crucial insights into e.g, ice dynamics, mass balance, and ancient landscapes buried beneath
5  the ice. The integration of radar data with other geophysical methods has enhanced bathymetric models, improving predictions of ice–ocean interactions and ice-shelf stability and contributed to a better understanding of crustal and geological evolution of the Antarctic continent. To support scientific progress, AWI made its airborne radar data publicly accessible through the *Radar Data over Polar Ice Sheets* viewer hosted by the Marine Data Portal (https://marine-data.de/viewers/) and PANGAEA
10  (https://doi.org/10.1594/PANGAEA.972094), ensuring compliance with FAIR principles. Future research will expand on these contributions, focusing on refining ice-sheet models and exploring new areas of glaciological and geological interest.



## 1 Introduction

The Greenland and Antarctic Ice Sheets are critical components of the Earth's climate system and play a vital role in regulating global sea levels. Together, these ice sheets hold approximately 99 % of the world's freshwater ice (Church et al., 2013), with their combined potential contribution to sea level rise estimated at over 65 m if fully melted (Morlighem et al., 2017, 2020). In recent decades, the Greenland Ice Sheet (GrIS) and the Antarctic Ice Sheet (AIS) have exhibited accelerated mass loss, driven by surface melting, ice dynamics, and interactions with warming oceans (Shepherd et al., 2020). Beyond their implications for sea level, the ice sheets exert significant influence on the global radiation budget, and on atmospheric and oceanic circulation patterns, affecting regional and global climate dynamics (Rahmstorf, 2007).

A detailed characterization of ice sheet structure and dynamics is essential to determine the mechanisms of mass loss and constrain numerical models of future behavior (Alley et al., 2019). Geophysical investigations provide powerful tools for studying the ice sheets, offering insights into their internal and basal properties, thickness, and interactions with the underlying bedrock. Airborne radar enables high-resolution imaging of ice thickness, ice-sheet stratigraphy, subglacial features as well as detecting changes in crystal orientation fabric (COF; e.g., Robin et al., 1969; Steinhage et al., 2001; Eisen et al., 2007; Bingham et al., 2015; Winter et al., 2019; Bodart et al., 2021; Gerber et al., 2023; Carter et al., 2024; Paxman et al., 2025). These data are critical for constraining ice-sheet models, which require accurate boundary conditions to simulate ice flow and predict the response of ice sheets to climatic and oceanic forcing.

Since 1994, the Alfred Wegener Institute, Helmholtz Centre for Polar and Marine Research (AWI) has operated airborne radar systems over Greenland and Antarctica (Figure 1) using AWI's polar aircraft (Figure 2; Alfred-Wegener-Institut, Helmholtz-Zentrum für Polar- und Meeresforschung, 2016b). The resulting archive of radar data constitutes a significant portion of the total global radar data archive for both ice sheets (Figure 1). In Antarctica, the data primarily cover East Antarctica's Dronning Maud Land, while in Greenland, most profiles cover the northern and northeastern parts of the ice sheet. The radar data were collected using six different systems (Figure 2 and Table 1), each designed to study various aspects of the ice sheet, offering different spatial resolutions and penetration depths (Figures 4, 5, and 6). AWI's radar data serve as a foundational dataset for numerous studies in glaciology, polar geology, and geodynamics.

This review paper aims to synthesize three decades of AWI's airborne radar surveys over the Antarctic and Greenland ice sheets, highlighting the technological advancements, scientific achievements, and collaborative efforts that have shaped our understanding of the polar ice sheets. We provide a comprehensive overview of AWI's radar systems, their technical specifications, and their diverse scientific applications that underscore the critical role these data have played in advancing glaciological, geological, and oceanographic research. The paper also marks the public release of AWI's airborne radar data over the polar ice sheets. We present the online viewer for *Radar Data Over Polar Ice Sheets* of the Marine Data Portal (https://marine-data.de/viewers), a web platform that allows users to explore the archive of AWI radar data. Finally, we describe how to access AWI's radar data products via the PANGAEA Data Publisher (https://doi.org/10.1594/PANGAEA.972094; Eisen et al., 2024) from where the data can be freely downloaded. Ultimately, this review not only focuses on the legacy of AWI's



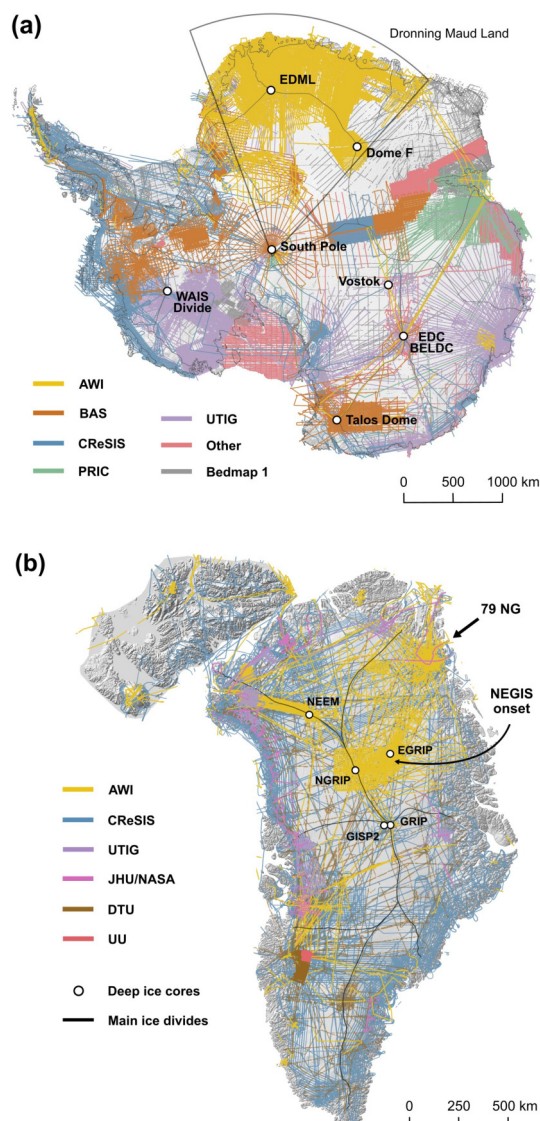

**Figure 1.** Institutes with substantial radar profile coverage in (a) Antarctica and (b) the Arctic: Alfred Wegener Institute (AWI), British Antarctic Survey (BAS), Centre for Remote Sensing and Integrated Systems (CReSIS), Polar Research Institute of China (PRIC), University of Texas Institute of Geophysics (UTIG), John Hopkins University (JHU), Techical University of Denmark (DTU), Upsalla University (UU), and other institutes (see Morlighem et al. 2017; Karlsson et al. 2024 for details in Greenland and Bingham et al. 2025 for details in Antarctica). For the AIS the radar data coverage of the Bedmap 1 period is shown in grey in the background. The black lines represent the main ice divides from Mouginot et al. (2017) and Rignot et al. (2019) and the white circles deep ice-core locations. Note that the spatial scale for Greenland is twice as large as for Antarctica. The two arrows in (b) highlight the Nioghalvfjerdsfjorden Glacier (79° NG) and the onset region of the Northeast Greenland Ice Stream (NEGIS).



contributions but also sets the stage for continued innovation in polar research, emphasizing the importance of open data sharing and interdisciplinary collaboration in addressing the challenges posed by climate change.

## 2  AWI airborne platforms and logistical infrastructure for radar surveys

Airborne radar surveys conducted by AWI have been carried out since 1994 using Polar aircraft of the types Dornier Do 228, and Basler BT-67 – a modern version of the Douglas DC-3 manufactured by Basler Turbo Conversions in Canada (Alfred-

Wegener-Institut, Helmholtz-Zentrum für Polar- und Meeresforschung, 2016b, Figure 2). The Basler BT-67 type has two engines, each with a power output of 955 kW, a range of approximately 3 000 km (depending on payload and survey design), a maximum take-off weight of around 13 t (depending on takeoff altitude), and a cruising speed up to 315 km/h.

Between 1994 and 2006, radar surveys were conducted with Polar 2 (Dornier Do228), since 2007 with Polar 5 and, from 2011 onward, also with Polar 6 (both Basler BT-67). All the aircraft are capable of operating on skis. The advantages of the

Basler type are its greater range and, most notably, significantly higher capacity for carrying scientific instruments and cargo. This capability allows for combined geophysical surveys using a variety of instruments, such as radar, gravimetry, magnetics, laser scanners, and more.

In addition to the aircraft platforms, AWI radar survey flights rely on extensive infrastructure in Antarctica and the Arctic, as well as in Bremerhaven and Bremen (Germany). In Antarctica, the three generations of the German Neumayer polar research

station and Kohnen Station (Alfred-Wegener-Institut, Helmholtz-Zentrum für Polar- und Meeresforschung, 2016a) have served as a logistical hub for AWI research expeditions since 1981. Neumayer Station is occupied year-round by overwintering staff since 1981 (Franke et al., 2022b), allowing for the use of a prepared airfield throughout the entire Antarctic summer season. With the opening of Kohnen Station at the EPICA Dronning Maud Land (EDML) ice-core drill site in 2001, the range of AWI airborne surveys was significantly extended into the interior of the East Antarctic plateau. Additionally, through cooperation

with other research stations, further airfields in Antarctica are utilized.

In Greenland, existing commercial or military airport infrastructure has been used, as well as permanent or temporary inland bases. The temporary bases were primarily associated with ice-core deep drilling projects, such as at NorthGRIP, NEEM, and EastGRIP, from which several extensive surveys were conducted.

The infrastructure network is complemented in Germany by logistics and coordination in Bremerhaven and at an aircraft

hangar in Bremen (formerly in Bremerhaven), where equipment installations and from where test flights take place. The flight crew consists of engineers and researchers from AWI, along with pilots and mechanics from different contractors over time.

## 3  AWI airborne radar systems

The following section provides a detailed overview of the various radar systems used by AWI, describing both their technical specifications, data processing, and their main areas of deployment over the years. Technical parameters of the radar systems

are presented in Table 1. Example radargrams from the individual systems can be found in Figures 4, 5, and 6. An overview





of the workflow from data acquisition to archiving, analysis, and public availability is illustrated in the flow chart diagram in Figure 7. The coverage of radar profiles for each radar system is shown in Figures 8 and 9 (organized by radar system) and in Appendix Figures A1 and A2, as well as Appendix Tables A1 and A2 (organized by season and campaign).

For each system we first describe the technical specification followed by the applied processing steps for each system, respectively. Finally, we briefly describe where the radar systems were deployed in Antarctica and Greenland.

### 3.1 Electromagnetic Reflection System (EMR)

#### 3.1.1 Technical specifications

AWI's EMR (Electromagnetic Reflection) system was built in cooperation with Aerodata Flugmesstechnik GmbH, Technische Universität Hamburg-Harburg (Germany) and the German Aerospace Center (DLR) and has been operating in Antarctica and Greenland since 1994 (Nixdorf et al., 1999). The system comprises two short backfire antennas in bistatic mode, which are mounted underneath the wings of AWI's polar aircraft (Fig. 2). The transmission signal generation is controlled by a timing board based on an oscillator with a reference frequency of 10 MHz and allows transmission pulse lengths of between 60 and 600 ns with a pulse repetition frequency (PRF) of 20 kHz. Aircraft positioning is determined by an inertial navigation system (INS), barometric altimetry, global positioning system (GPS) and, since 1997, laser altimetry (Nixdorf et al., 1999). The EMR system records on three channels (two channels prior to 1998) with each data sample assigned to a channel based on signal strength (Nixdorf et al., 1999).

Since 1998, the EMR system has been able to operate in the "toggle mode", switching between transmission pulse lengths of 60 and 600 ns, respectively, to image internal layers at shallow and intermediate depths at high resolution ($\sim 5$ m range resolution) and the bed reflection in deep ice at low resolution ($\sim 50$ m range resolution). The along-track trace spacing depends on trace coordinate precision and along-track stacking during the processing and typically lies in the range $\sim 50 - 100$ m. In pre-1998 surveys, some EMR datasets may consist of a combination of short- and long-pulse data. The profile is then classified as either a short- or long-pulse profile, depending on which type of data predominates in the profile.

Over the three decades of acquisition, three different data acquisition systems were used, which we present in the following subsections.

**Aerodata System (1994 – 2004)**

Between 1994 and 2004, EMR data were acquired using the Aerodata system with Modams radar data acquisition system. Measurements were taken with a fixed time window of 50 $\mu$s, a sampling interval of 13.33 ns, a PRF of 20 kHz, and a data rate of 20 Hz. The data were vertically stacked by a fixed factor of 200. An exception is two flights during the Arctic season of 1998, which were measured with a PRF of 40 kHz. In 1997, the initial two-channel system was extended by a three-channel system, yielding better resolution for weaker signals.





**Optimare System (2004 – 2012)**

Between 2004 and 2012, the Optimare system with Medusa P data acquisition was used. As with the Aerodata system, a time window of 50 $\mu$s, a sampling interval of 13.33 ns, and a PRF of 20 kHz were employed. However, the vertical stacking factor was increased to 992, and a three-channel system was consistently utilized.

**WERUM System (since 2012)**

Since 2012, the WERUM system with ADA data acquisition has been used for EMR measurements. With the Werum system, the transmitter was modified so that the transmission signal is always generated with the same phase. The recording time window is 50 $\mu$s, the sampling interval is 4 ns, the data rate is 20 MHz, and the PRF is 20 kHz. The stacking is freely adjustable but is typically set to a factor of 1000. In addition to this configuration, three further modifications have been made to the

system in this period: (i) a logarithmic detector (logdet) was installed; (ii) the same logdet was used, but with a time window of 64 $\mu$s, a PRF of 15 kHz, and a data rate of 15 Hz; and (iii) a new logdet was installed and used with the same settings as in (ii).

### 3.1.2 Deployments of the EMR System since 1994

The EMR system has been primarily deployed in Antarctica's Dronning Maud Land, particularly around the EDML ice-core

deep drilling site and in the areas both south and north of the coastal escarpment to map ice thickness and englacial stratigraphy. Several campaigns also focused on connecting East Antarctic ice cores, such as between EDML and Dome Fuji and EDML and the South Pole, as well as between Vostok, EDC, and Talos Dome. Furthermore, EMR RES surveys extend east and west beyond Dronning Maud Land, and cover parts of the Ronne ice shelf and the Antarctic Peninsula.

The EMR system has been extensively used in Greenland, primarily in the northeast where it covers all major deep ice core

drilling sites (e.g., the Greenland Ice Core Project GRIP). Some surveys extend towards the northwest and southwest. In 1996, the NorthGRIP drilling site was covered by an extensive grid survey.



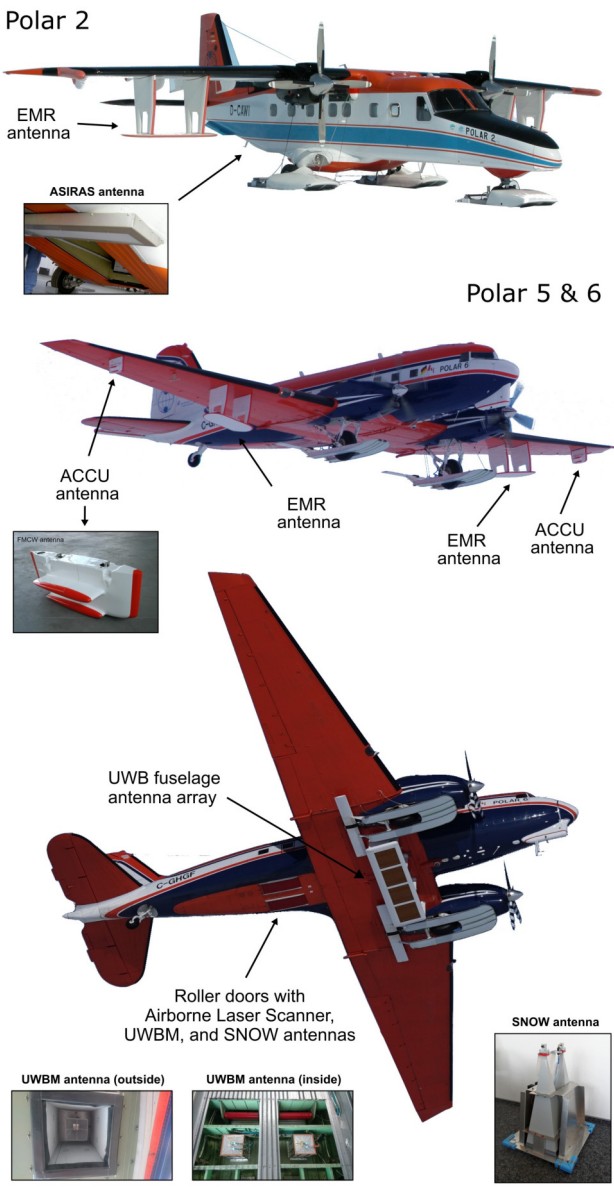

**Figure 2.** AWI's polar aircraft Polar 2 (Dornier Do228; registration code: D-CAWI), Polar 5 (Basler BT-67; registration code: C-GAWI), and Polar 6 (Basler BT-67; registration code: C-G HGF) and the six AWI radar systems. The bottommost picture of the airplane shows the fuselage antenna of the UWB system, without wing antennas mounted. The bottommost photo of the Basler aircraft was taken by Sepp Kipfstuhl (AWI). All other photos were taken by the authors of this manuscript.





**Table 1.** AWI radar system specifications. Abbreviations: TU HH = Technical University Hamburg-Harburg; RST = Radar Systemtechnik; CReSIS = Centre for Remote Sensing and Integrated Systems; MCoRDS = Multi-channel Coherent Radar Depth Sounder; FMCW = Frequency-Modulated-Continuous-Wave; IRH = Internal Reflection Horizon.

| Radar system | Acronym | Manufacturer | Frequency | Transmit signal | Range resolution | Scientific focus |
|---|---|---|---|---|---|---|
| Electromagnetic Reflection system | EMR | TU HH | 150 MHz | 600 ns burst<br>60 ns burst | 50 m<br>5 m | IRHs,<br>ice thickness |
| Accumulation radar | ACCU | TU HH | $400-800$ MHz[1] | FMCW chirp | 0.2 m[1] | Firn & IRHs |
| Snow radar | SNOW | TU HH | $8-12$ GHz | FMCW chirp | $\sim 2$ cm | Snow & firn |
| Airborne SAR/Interfer. Radar Altimeter System | ASIRAS | RST | $13-14.5$ GHz | $5-45\,\mu s$ & $80\,\mu s$ chirp[2] | $\sim 7$ cm | Snow & firn |
| Ultra-Wideband radar (MCoRDS 5) | UWB | CReSIS | $150-600$ MHz | $1-10\,\mu s$ chirp | $0.3-5$ m | IRHs,<br>ice thickness,<br>englacial features,<br>swath bed imaging |
| Ultra-Wideband microwave radar | UWBM | CReSIS | $2-18$ GHz | $240\,\mu s$ chirp | $\sim 1$ cm | Snow & firn |

[1] During the ARK 2010 campaign a bandwidth of $500-700$ MHz was used and the range resolution was 0.5 m.

[2] In the high-altitude mode the ASITAS system can operate in a linear frequency modulated chirp length between $5-45\,\mu s$. The low-altitude mode has a chirp length of $80\,\mu s$.

## 3.2 Accumulation radar (ACCU)

### 3.2.1 Technical specifications

The airborne Accumulation (ACCU) radar system is a Frequency Modulated Continuous Wave (FMCW) radar system op-
erating in a frequency range of $500-700$ MHz in the Arctic season in 2010 (Jenett and Steinhage, 2012) and was increased thereafter to $400-800$ MHz. It is capable of detecting internal structure in the upper 200 m of the ice sheets with a vertical resolution of 1 m or better (Fig. 4d and 5a). Moreover, the ACCU system can detect the ice-shelf bottom down to 400 m depth or more. The trace spacing of the ACCU system depends on the trace coordinate accuracy and varies between $10-40$ m. The radar was designed primarily to map shallow internal reflection horizons (IRHs) in Antarctica and Greenland.

### 3.2.2 Deployments of the ACCU system since 2010

Initial airborne measurements using this system were conducted in 2010 in Greenland during the NEEM campaign in north-west Greenland with Polar 5. In addition to the ACCU FMCW system, Polar 5 was complementarily equipped with two laser altimeters, ESA's ASIRAS radar altimeter, and the EMR to measure ice thickness. In 2012, the ACCU radar was used to map central Greenland's near-surface stratigraphy between the NEEM, NorthGRIP and EastGRIP deep ice core sites.

Between the Antarctic seasons 2011/12 and 2016/17 the ACCU radar was deployed in Antarctica's Dronning Maud Land running in parallel with the EMR and SNOW systems in the 2013/14 season.



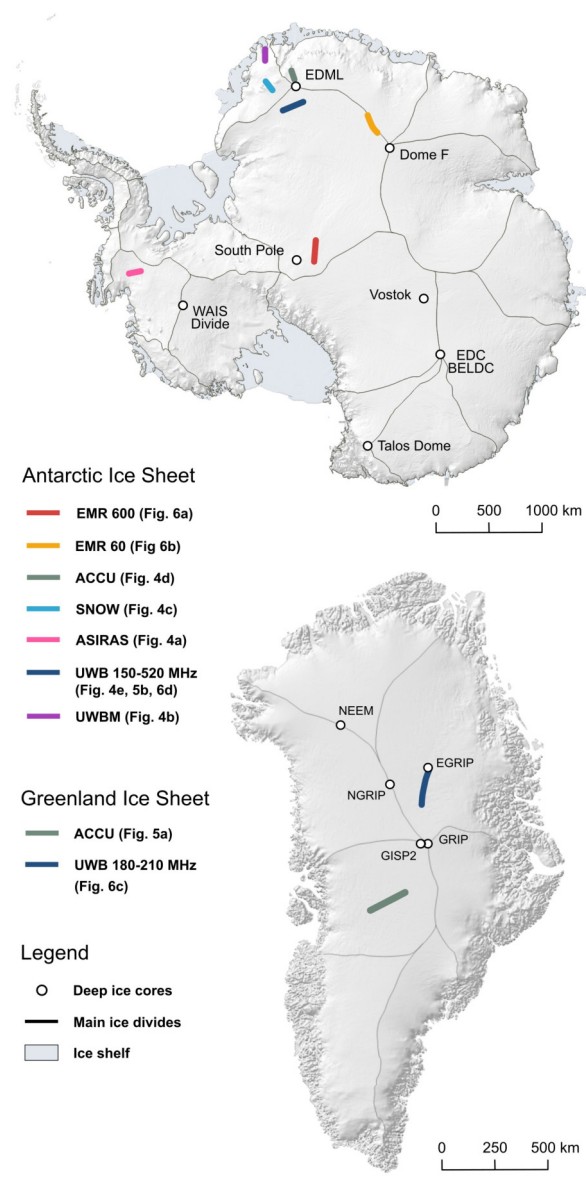

**Figure 3.** Profile locations of radargram examples in Figures 4, 5, and 6 over the Antarctic and Greenland Ice sheets.

## 3.3 Snow radar (SNOW)

### 3.3.1 Technical specifications

The snow radar system (SNOW) is a FMCW system transmitting in a frequency range of 8 – 12 GHz. It was developed by the

TU Hamburg-Harburg (Germany). The system operates with a transmit power of up to 3.2 W and a sampling rate of 500 MHz.





**Figure 4.** Examples of radargrams focusing on the first 50 m depth below the surface. (a) ASIRAS; profile ID: ASR_L1B_01_20141203T130418_132058, (b) UWBM; profile ID: UWBM_20221205_99_[019-021], (c) SNOW; profile ID: SNOW_20140102T074743_081922, (d) ACCU; profile ID: ACCU_20131230T090855_094516, (e) UWB in wideband mode; profile ID: Data_20231211_02_006_standard. For TWT-to-depth (two-way travel time) conversion we used an average electromagnetic wave speed in the upper 50 m of $2 \cdot 10^9$ m s$^{-1}$, which corresponds to an average ordinary relative dielectric permittivity $\varepsilon_r'$ of 2.25. The location of the radar profile is annotated in the lower left corner of the radargram. All radargrams are time-zeroed at the ice surface reflection.



**(a)**



**Figure 5.** Examples of radargrams focusing on the first $\sim 175\,\text{m}$ depth below the surface. (a) ACCU; profile ID: ACCU_20100722T132929_140949, (b) UWB in wideband mode; profile ID: Data_20231211_02_008–010_standard. For TWT-to-depth (two-way travel time) conversion we used an average electromagnetic wave speed in the upper $175\,\text{m}$ of $1.75 \cdot 10^9\,\text{m}\,\text{s}^{-1}$, which corresponds to an average ordinary relative dielectric permittivity $\varepsilon_r'$ of 3.0. The location of the radar profile is indicated in the lower left corner of the radargram. All radargrams are time-zeroed at the ice surface reflection.

The penetration depth is limited to the upper $10-20\,\text{m}$ (Figure 4c) with decreasing signal to noise ratio with depth. The theoretical range resolution is $\sim 2\,\text{cm}$ and trace spacing depends on the trace coordinate accuracy and varies between 10 and $40\,\text{m}$.

### 3.3.2 Deployment of the SNOW system

The SNOW radar was used as an airborne system over land ice during the Antarctic season of 2013/14. Its profiles are centered on the EDML ice core and extend radially north and west towards the coast, south towards the Bailey and Slessor Ice Streams,







**Figure 6.** Examples of deep-sounding radargrams. (a) EMR long-pulse; profile ID: 20112120, (b) EMR short-pulse; profile ID: 20033141, (c) UWB narrowband; profile ID: Data_20180510_01_013_standard, (d) UWB wideband; profile ID: Data_20231211_02_006–008_standard. For TWT-to-depth conversion we used an average electromagnetic wave speed of $1.689 \cdot 10^9 \, \mathrm{m \, s^{-1}}$, which corresponds to an average ordinary relative dielectric permittivity $\varepsilon'_r$ of 3.15. The location of the radar profile is indicated in the lower left corner of the radargrams. All radargrams are time-zeroed at the ice surface reflection.

and east along the ice divide between EDML and Dome Fuji (Fig. 8e). In addition, the SNOW system also operated over sea ice during the ANT 2014/15, ARK 2014, and ARK 2015 seasons.





## 3.4 Airborne SAR/Interferometric Radar Altimeter System (ASIRAS)

### 3.4.1 Technical specifications

The Airborne SAR/Interferometric Radar Altimeter System (ASIRAS) is a radar system developed by ESA as part of the CryoSat mission (Lentz et al., 2002). ASIRAS was built by the Swiss company Radar Systemtechnik (RST). Support was provided by AWI as well as the company Optimare for the integration and operation of the device in the research aircraft.

ASIRAS was designed to demonstrate the concept of a SAR-interferometric altimeter, providing support for CryoSat's 160 mission to measure polar ice thickness (Mavrocordatos et al., 2004) within the framework of the CryoSat Validation Experiment (CryoVEx). The system was developed to support ESA's CryoSat mission by providing airborne measurements that could be used to calibrate, validate and refine satellite data for ice and snow cover, helping improve the understanding of polar regions.

ASIRAS is one of ESA's first new-generation radar altimeters that employ pulse-width limited technology for higher precision. It operates similarly to a Ku-band altimeter, using a carrier frequency of 13.5 GHz and a bandwidth of 1 GHz. The range 165 resolution of the system is $\sim 7$ cm. The high pulse repetition frequency and pulse to pulse coherence allows delay/Doppler unfocused SAR processing (Raney, 1998) and thus a fine along-track resolution of $< 5$ m to estimate highly detailed, accurate surface elevation data. ASIRAS also includes a dual-antenna system positioned across-track, functioning as a single-pass interferometer to enhance spatial resolution (Lentz et al., 2002; Hawley et al., 2006).

### 3.4.2 Data processing

The ASIRAS data processing follows the range-Doppler principle described by Raney (1998). ESA provides the processing software, allowing network-wide processing of ASIRAS data. Over time, numerous corrections and updates were made to the software based on joint testing by AWI and ESA, and these updates were integrated into the system (Helm, 2008).

The processing is divided into two stages: Level 1 and Level 1B (Helm, 2008). Level 1 georeferences radar bursts using interpolated GPS and INS data and applies corrections for sensor rotations. Its outputs feed into Level 1B processing, which 175 transforms bursts into the range-Doppler domain via fast Fourier transformations, applies corrections like Doppler centroid and slant range adjustments, and enhances the signal to noise ratio through beam stacking. The final Level 1B product includes georeferenced radar echoes, preliminary terrain heights from an Offset Center Of Gravity (OCOG) retracker, and additional metrics such as power and phase. These data are stored in binary format and converted into FAIR user-accessible data products (Figure 7).

### 3.4.3 Deployments of the ASIRAS system between 2004 and 2019

ASIRAS has been successfully flown on several measurement campaigns mostly by the CryoSat Validation and Retrieval Team. The system was deployed across significant portions of Greenland and the Canadian Arctic, with flights between 2004 and 2019 (Figure 9f), where other radar systems of AWI were also used. Except for the 2004 campaign, none of these Arctic ASIRAS system flights used AWI aircraft. Instead, they were conducted by ESA and DTU (Technical University of Denmark).




Hence, the individual flight seasons are not detailed in this paper. However, as the radar data from the ASIRAS system were processed at AWI and the radar data products are provided through AWI, they are included in the overview maps (Figures 8g and 9f). Additionally, between 2007 and 2018, numerous flights were conducted across extensive parts of Antarctica (the majority with AWI polar aircraft), primarily focusing on the coastal regions (Figure 8g).

## 3.5 Ultra-Wideband radar (UWB)

### 3.5.1 Technical specifications

AWI's ultra-wideband (UWB) airborne radar system is an improved version of the Multichannel Coherent Radar Depth Sounder (MCoRDS, version 5), which was developed at the Center for Remote Sensing of Integrated Systems (CReSIS) at the University of Kansas (Rodriguez-Morales et al., 2013; Hale et al., 2016). It has an improved hardware design compared to CReSIS' predecessor radar depth sounders (Gogineni et al., 1998; Wang et al., 2016). The basic radar configuration consists

of an eight element antenna array mounted under AWI's Polar 5 or Polar 6 Basler BT-67 aircraft's fuselage (Figure 2). The fuselage antenna elements function as transmit and receive channels using a transmit–receive switch. Additionally, two eight element receiver arrays can be mounted underneath the wings to increase the signal to noise ratio. This 24-channel configuration has so far only been used in 2016, in Northwest Greenland, for the Hiawatha survey (Kjær et al., 2018), and during test flights in the Antarctic 2016/17 season. The total transmit power is 6 kW, and the radar can be operated within the frequency

range of 150–600 MHz. The PRF is 10 kHz, and the sampling frequency is 1.6 GHz. The characteristics of the transmission signals as well as the recording settings can be freely programmed to enable higher dynamic range. Usually the transmission signal is composed of staged linear modulated chirp signal of $2-5$ waveforms (for instance 1 $\mu$s unamplified, 3 $\mu$s high-gain and 10 $\mu$s high-gain), which provides high-resolution imaging of different parts of the ice sheet. The position and orientation of the aircraft is determined by four NovAtel DL-V3 GPS receivers, which sample at 20 Hz. The GPS system operates with

dual-frequency tracking so that the position accuracy can be enhanced during post-processing. The range resolution of the UWB data products depend mainly on the chosen bandwidth and the along-track spacing typically ranges between 5 and 15 m.

### 3.5.2 UWB data processing

**2D processing (sounding mode)**

Standard processing techniques are performed with the OPR Toolbox (Open Polar Radar Toolbox; formerly termed CReSIS

Toolbox; Open Polar Radar, 2023). The main steps comprise motion compensation, pulse compression, synthetic aperture radar (SAR) focusing and array processing. The SAR processing is based on the fk (frequency–wavenumber) migration technique for layered sediment packages (Gazdag, 1978), which was adapted for radioglaciology (Leuschen et al., 2000). GPS data are post-processed by precise point positioning, with a final estimated accuracy (commercial software package Waypoint 8.4) of better than 3 cm for latitude and longitude, and better than 10 cm for altitude. For further details on radar data acquisition and

processing, see Rodriguez-Morales et al. (2013), Hale et al. (2016), and Franke et al. (2022c). When combining the different



images that highlight various depth ranges of the radargram, we also apply an amplitude correction to the individual images to reduce offsets at the image transitions. Moreover, we processed the data in a way that the TWT vector starts at 0 s.

**3D processing (swath/imaging mode)**

Swath processing is applied to produce a high-resolution digital elevation model of the bed topography. The information
from side-transmitting off-nadir returns from sequential acquisitions, combined with phase differences in arrivals between receiver elements (Jezek et al., 2011; Holschuh et al., 2020), are used to estimate the direction of arrival for energy in both the along-track and across-track directions (Paden et al., 2010). These data are SAR processed and combined channel by channel, using the Multiple Signal Classification (MUSIC) algorithm in the OPR Toolbox (Open Polar Radar, 2023), for mapping of the subglacial topography in three dimensions along a single flight line. Moreover, along-track and fast-time averaging was
applied to the SAR-processed data to enhance the signal to noise ratio for tracking the bed reflections and cross-track surface.

Bed elevation values from an existing bed topography DEM can be used to define the depth range for bed return, and together with the nadir bed return they serve as seed points for tracking cross-track reflectors. A Gaussian fit is applied to the bed return power to address broad distributions and multiple peaks, and tracking is constrained using a guided window based on a theoretical flat-bed hyperbola, minimizing interference from nearby englacial reflections. After tracking the cross-
track surface, a static angle correction accounts for surface tilting (if needed), and the bed return hyperbola is range migrated, including for air-to-ice refraction, to convert surface tracking to depth (Carter et al., 2024). Artifacts from upwarping at swath edges affect accuracy due to energy spreading. The range-migrated data are then converted into a point cloud and projected to geographic coordinates (latitude and longitude) using the true heading and flight trajectory from the radar data.

### 3.5.3 Deployments of the UWB system since 2016

The UWB system became fully operational in Greenland in 2016, deployed in northwest Greenland at the Hiawatha Glacier (Kjær et al., 2018) and in southwest Greenland at Jakobshavn Glacier. During this deployment, the system was flown in its 24-channel configuration with additional wing mounted arrays. This setup improved the signal to noise ratio but reduced the aircraft range. From 2018 onward, the UWB system was only used with a single array mounted under the fuselage of the Polar aircraft to increase the range of survey flights, as the surface clutter reduction offered by the 24-channel configuration was
deemed less valuable than the extended survey range possible without it.

Between 2018 and 2022, UWB flights in Greenland focused on the area around the EastGRIP drilling site (e.g., Franke et al., 2022c) and the upstream region of the 79° N glacier (79° NG) (Zeising et al., 2024). In 2018 and 2022, swath flights were conducted in the immediate vicinity of the EastGRIP ice core, producing a high-resolution (25 m horizontal resolution) DEM of the bed topography and landforms (Carter et al., 2025). In the 2022 survey, radar profiles were collected in full polarimetric
mode around NEGIS (Eisen et al., 2025) providing insights into englacial properties of this ice stream at comparable data quality of local ground-based polarimetric surveys in this region (Gerber et al., 2025; Nymand et al., 2025). Furthermore, in 2023, a campaign took place in the Canadian Arctic on the Müller Ice Cap to explore a suitable drilling site (Lilien et al., 2024) and in 2024, flights were conducted both upstream of the 79° NG and in Southwest Greenland.



The first deployment in Antarctica was conducted during the 2016/17 season, primarily consisting of test flights. In the 2018/19 season, surveys focused on the onset of the Jutulstraumen Glacier (e.g., Franke et al., 2021a, 2025b) as well as the area around the grounding line of the Roi Baudouin Ice Shelf and nearby Derwael Ice Rise (e.g., Koch et al., 2023; Zhou et al., 2025). During the 2023/24 and 2024/25 seasons, additional measurement flights were conducted directly at the grounding line in Dronning Maud Land, their aims led by the goals of the SCAR Action Group RINGS (RINGS Action Group, 2022), as well as flights mapping deep englacial stratigraphy (Franke et al., 2025b) and near-surface stratigraphy (Zuhr et al., 2025).

### 3.6 Ultra-Wideband Microwave radar (UWBM)

#### 3.6.1 Technical specifications

The ultra-wideband microwave radar (UWBM), is a $2 - 18$ GHz airborne FMCW radar developed by CReSIS at the University of Kansas (Yan et al., 2017a, b; Arnold et al., 2020). The radar system uses a chirp generator based on a direct digital synthesizer (DDS) with a frequency multiplier and a down-converter, dual-polarized transmitter and receiver antennae, intermediate frequency section, and a digital acquisition unit. The radar has four channels with full polarimetric transmit and recording capability (VV, HH: co-polarized, VH, and HV: cross-polarized). The radar transmits with a pulse length of $240\,\mu$s at an effective PRF of approximately 3.9 kHz and records at a sampling frequency of 125 MHz. The approximate range resolution is $\sim 1$ cm and the typical along-track spacing 1.35 m.

#### 3.6.2 Data processing

UWBM data undergo extensive processing, which involves removing coherent noise, often visible as undulating lines in radargrams (Fig. 4b), using a low-pass boxcar filter. The system far-field response is extracted using Discrete Fourier Transform methods to detect strong reflectors (e.g., open water leads) that provide high signal to noise ratio targets. The impulse response from these reflectors is used to deconvolve the radar signal, enhancing range resolution and minimizing sidelobes (Yan et al., 2017a).

Furthermore, the processing includes post-processing steps that perform motion compensation using the on-board GPS data and filter out erroneous data based on aircraft altitude and pitch/roll parameters, snow depth constraints, and surface temperature readings. Validation exercises over landfast sea ice show that this radar processing achieves a mean bias below 1 cm (Jutila et al., 2022), demonstrating the reliability of the UWBM radar system for high-resolution snow depth measurement on Arctic sea ice (Jutila et al., 2021, 2022; Franke et al., 2025a) and grounded ice (Humbert et al., 2020).

#### 3.6.3 Deployments of the UWBM system since 2016

After a testing phase in $2015 - 2016$, the UWBM was first deployed on land ice alongside the UWB in the 2016 season in northwest and northeast Greenland. A second deployment of the same instrument combination was carried out in the 2018 Greenland season to map near-surface features (e.g., Humbert et al., 2020). The UWBM has also been used in the Arctic for several campaigns since 2017 over sea ice, in combination with other instruments such as the airborne laser scanner and EM



Bird, as part of the AWI Icebird campaigns (Jutila et al., 2021, 2022). Since this article focuses on radar survey flights over land ice and ice shelves, the flights over sea ice are not included here.

During the Antarctic season 2022/23, the first UWBM deployment on Antarctic land ice took place in coastal western Dronning Maud Land. Data were collected near Neumayer Station over the Halvarryggen ice rise, and along the traverse between Neumayer and Kohnen stations, as well as on the fast ice of Atka Bay. In Atka Bay, the UWBM was able to map snow
thickness and detected flooded sea ice regions on the iceberg-laden sea ice (**?**).

## 4 AWI radar workflow

### 4.1 Data acquisition, processing, format conversion and archiving

For the various airborne radar systems operated by AWI, data acquisition and processing rely on different hardware and software packages. These differing workflows converge into a unified framework after the creation of processed radar data
products. In this way, standardized data formats are made available from all radar systems for end users, enabling efficient usage, broad accessibility, and consistent archiving. The fundamental workflow or data flow is illustrated in Figure 7.

At the acquisition level of each radar system, data undergo preliminary processing before being written to storage. This step may include horizontal and vertical stacking or signal amplification. The raw data at acquisition are stored as primary data in the AWI tape archive within a protected, write-secure area. At the processing level, the raw data are input to various
software packages, e.g., the OPR Toolbox (formerly CReSIS Toolbox; Open Polar Radar, 2023) for UWB and UWBM data, the FMCW Converter for ASIRAS, ACCU, and SNOW data, and the software Paradigm™ from AspenTech Subsurface Science & Engineering for EMR data. The processing outputs, depending on the software, are saved as primary data products in a dedicated section of the AWI tape archive designed for long-term storage, offering high security against data loss.

To ensure uniform data formats and simplified usability, the processed data products in their diverse formats are integrated
into an IDL Data Converter. The output consists of standardized netCDF files for radar data, geodata such as KML and shapefiles, quicklook images of radargrams, and standardized SEGY files and coordinate files for import into commercial seismic software (e.g. Paradigm™) for post-processing and layer picking. These converted files are also stored on the AWI Hybrid-NAS-Storage, which offers fast connectivity to the internal network and high performance computing resources for analysis and post-processing.
Furthermore, the standardized radar data are made publicly available at two central repositories. Radar data (netCDF files), radar profile coordinates (KML files), and quicklook radargram images (JPGs) are archived on PANGAEA (Felden et al., 2023; Eisen et al., 2024). Additionally, radar profile coordinates, quicklook radargrams, and other metadata are forwarded to the Marine Data Portal.



**(a)**



**Figure 7.** Flow chart of AWI radar data from (a) acquisition, over (b) processing and (c) conversion, to (d) public data access and post-processing. The panels on the right represent the level of data backup for the respective data level.

## 4.2 Ice surface reflection determination

To determine the surface reflection, we use a leading-edge retracker for all radar data products. This retracker applies the TCOG (Threshold Centre of Gravity) algorithm (Davis, 1997) within a specific window, capturing the leading edge with a threshold of 0.8. This approach provides a consistent method for determining surface reflection across all radar products by identifying it at the gradient's ascent toward the maximum.





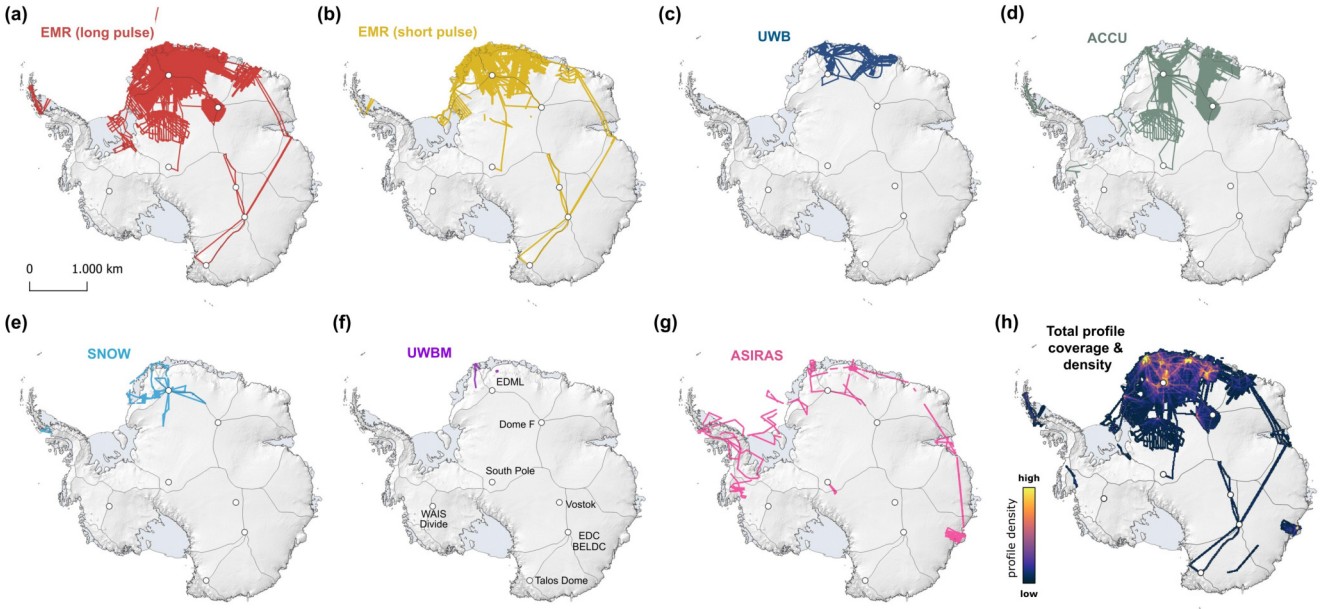

**Figure 8.** AWI radar data coverage over the Antarctic Ice Sheet categorized by radar systems. (a) Electromagnetic Reflection system (EMR) long-pulse (600 ns), (b) EMR short-pulse (60 ns), (c) Ultra-wideband (UWB) radar, (d) Accumulation Radar (ACCU), (e) Snow radar (SNOW), and (f) Ultra-wideband microwave (UWBM) radar, (g) Airborne SAR/Interferometric Radar Altimeter System (ASIRAS), (h) spatial profile density of all AWI radar systems. For the computation of the spatial profile density, we exclude the ASIRAS profiles. The white circles represent deep ice core locations and the black lines the IMBIE drainage basins (Rignot et al., 2019).

We choose this approach instead of allocating surface reflection at the local maximum because the maximum can be broad,
saturated, and potentially characterized by multiple peaks. In a few cases, the algorithm fails: for example, when sidelobes near the surface are too strong, when the tracking window migrates out of the surface reflection due to significant changes in flight altitude, when strong near-surface reflections occur, or when aircraft roll is large. These cases were manually corrected for EMR, UWB and UWBM data.

## 5  AWI radar surveys

So far, AWI has collected more than 1 610 000 profile-km of radar data, i.e., the sum of the length of all radar products from the different radar systems. This data have been used for a wide range of scientific objectives covering a variety of glaciological and geophysical problems in the polar regions. In this section, we present a summary of key scientific targets and outcomes.



## 5.1 Ice thickness surveys in East Antarctica and Greenland

Since 1994, AWI has conducted extensive radar surveys to determine the ice thickness of the Antarctic and Greenland ice
sheets. In Antarctica, these surveys focused mainly on Dronning Maud Land (Steinhage et al., 1999, 2001; Riedel et al., 2012;
Karlsson et al., 2018; Eisermann et al., 2020, 2021; Franke et al., 2021a). They have been contributing a considerable amount
of data to continent-wide ice thickness and bed topography datasets such as Bedmap (Lythe and Vaughan, 2001; Fretwell
et al., 2013; Frémand et al., 2023; Pritchard et al., 2025) and BedMachine Antarctica (Morlighem et al., 2020) as well as for
modelling studies (e.g., Kleiner and Humbert, 2014, see Section 5.10 for details). In Greenland, the measured ice thickness
primarily covers the northeastern part of the ice sheet (Mayer et al., 1999; Franke et al., 2020, 2022a; Zeising et al., 2024)
and the area surrounding the NorthGRIP ice core (Nixdorf and Göktas, 2001). The bed topography derived from a survey
with the UWB system led to the discovery of a large impact crater beneath Hiawatha Glacier in northwest Greenland (Kjær
et al., 2018). All ice thickness data together form an essential component of Greenland-wide ice thickness and bed topography
products (Bamber et al., 2013; Morlighem et al., 2017).

The majority of ice thickness measurements, both in Greenland and Antarctica, were obtained from surveys with the EMR
radar, which, however, was largely replaced by surveys using the UWB radar since 2016 for glaciological objectives and
projects. For aerogeophysical surveys, the simpler and lighter EMR system remains a standard system for combination with
gravity and magnetic sensors. Testing of a lighter gravimeter alongside EMR in 2022/23 (Johann et al., 2025) paved the way for
gravimetry together with UWB for the first time in the Antarctic season of 2024/25. Despite the weight saving, this combination
still offers a shorter range than is possible with the EMR system.

Both radar systems are capable of sounding ice more than three kilometers thick, but the improved configuration and pro-
cessing capabilities of the UWB system provide better range and along-track resolution of the bed topography. For ice thickness
determinations using the EMR radar, long-pulse data were used to identify the base reflection in profiles where (i) short-pulse
data were not available or (ii) the base reflection was not visible in the short-pulse data. Since short-pulse data offer better
vertical resolution, they were preferred for ice thickness determination where the base reflection is visible due to the higher
vertical resolution of the data product.

## 5.2 Basal properties

In addition to studies focusing on ice-thickness determination, AWI radar data have also been used to examine the detailed
properties of the ice sheet base. In both Greenland and Antarctica, bed topography analyses have been employed to investigate
valley structures and the roughness of the bed using various metrics (Eisen et al., 2020; Franke et al., 2021b, a). These studies
range from a contribution of data in DML to continental-scale studies (Eisen et al., 2020), to local and regional contexts to
explore the relationship between ice flow, its direction, and the direction-dependent basal roughness (Franke et al., 2021b) as
well as landscape preservation and origin (Franke et al., 2021a). To further refine our understanding of bedrock roughness on
smaller spatial scales, parameters like the abruptness of the base reflection and characteristic side reflections parallel to flight
tracks have been analyzed. For instance at the onset region of the Northeast Greenland Ice Stream (NEGIS) off-nadir reflections





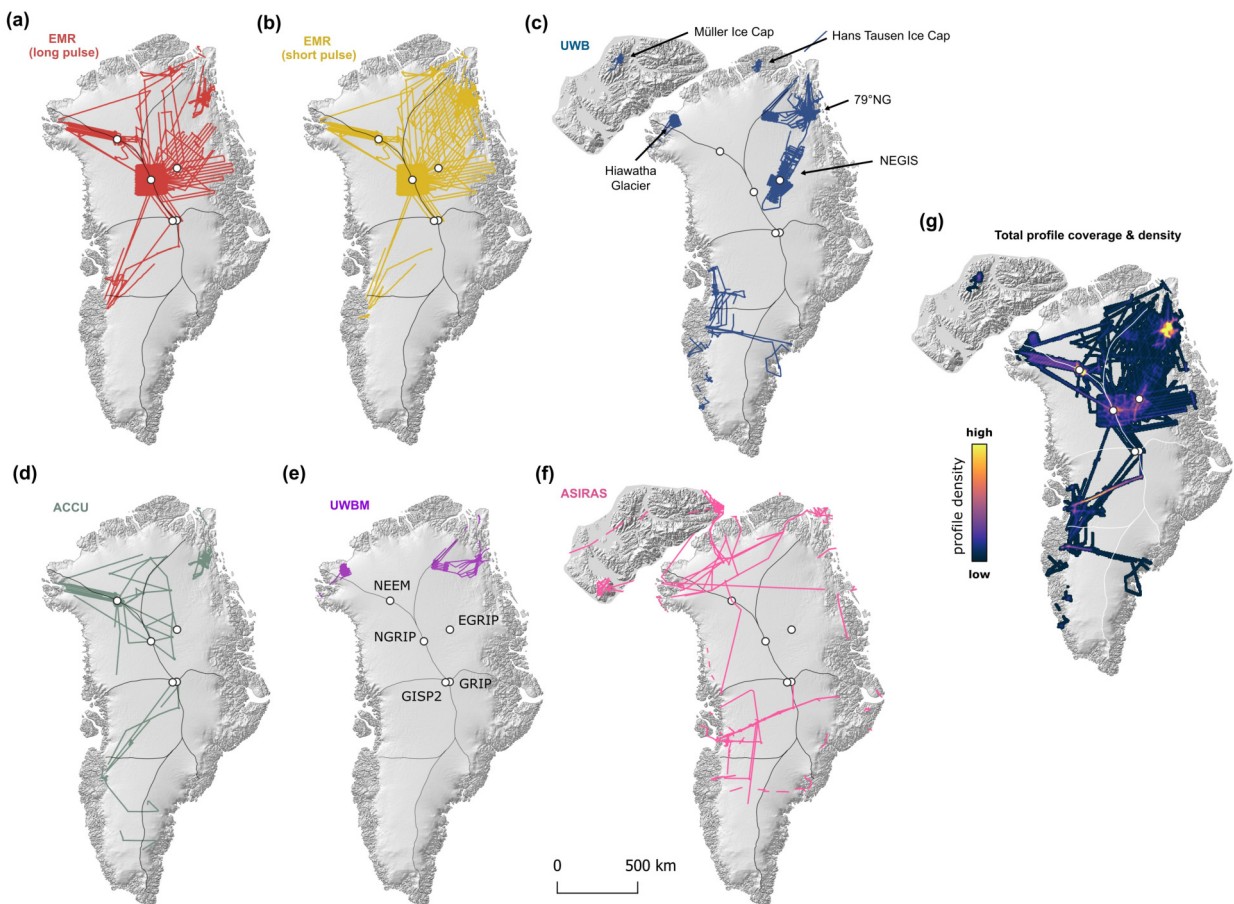

**Figure 9.** AWI radar data coverage over the Greenland Ice Sheet and Canadian Arctic categorized by radar systems. (a) Electromagnetic Reflection System (EMR) long-pulse (600 ns) profiles, (b) EMR short-pulse (60 ns) data, (c) Ultra-wideband (UWB) radar data, (d) Accumulation Radar (ACCUM) data, (e) Ultra-wideband microwave (UWBM) radar data, (f) Airborne SAR/Interferometric Radar Altimeter System (ASIRAS), (g) spatial profile density of all radar surveys. For the computation of the spatial profile density, we exclude the ASIRAS profiles. The white circles represent deep ice core locations and the black lines the IMBIE drainage basins (Mouginot et al., 2017).

indicate the presence of elongated subglacial landforms oriented parallel to ice flow and shaped by ice stream activity (Franke et al., 2020, 2021b), and were categorized in a high-resolution DEM of the bed topography as mega-scale glacial lineations (Carter et al., 2025).

Next to mapping bedrock topography and subglacial landscapes shaped by glacial activity, radar data are also important for analyzing the transport and storage of subglacial water. High-resolution ice thickness measurements are essential for modelling how subglacial water moves and identifying potential locations where it might pool, such as subglacial lakes (Goeller et al., 2016). While radar data alone are often not sufficient for unambiguous lake detection, they serve as a crucial foundation for combination with other measurements or models. For example, in western DML, radar data combined with SAR interferometry



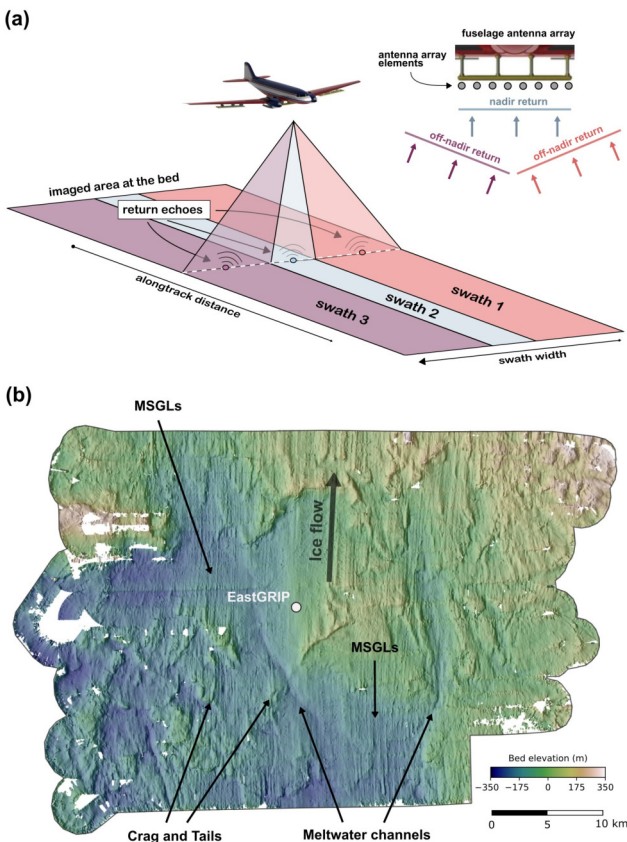

**Figure 10.** AWI UWB swath radar capability to image high-resolution bed topography. (a) Sketch of the swath radar aquisition principle. (b) Digital elevation model of the bed topography around the EastGRIP drill site at the onset region of the NEGIS with a grid resolution of 25 m. The DEM highlights subglacial landforms, such as mega-scale glacial lineations (MSGLs), crag-and-tails, and basal meltwater channels (Carter et al., 2025). Both figures are modified from Carter et al. (2025).

have shown how subglacial water moves between topographic depressions (Neckel et al., 2021). Additionally, studies at the

onset of the Recovery Ice Stream have raised doubts about whether subglacial water exists in large lakes, challenging previous assumptions (Humbert et al., 2018). However, it has been shown that the interpretation of reflections at locations where active lakes have been reported to be filled, can be inconclusive (Hills et al., 2024)

In addition to radar signals at the ice–base interface, the reflection characteristics of basal ice are also valuable for understanding basal properties and processes. High-resolution ultra-wideband (UWB) data from a 2018 survey of Jutulstraumen

Glacier revealed embedded point scatterers, which are likely sediment particles instead of water pockets (Franke et al., 2023b). Sediments are likely to become entrained when subglacial water freezes to the ice base, influencing not only the mechanical properties of the basal ice during flow but also providing insights into the basal temperature regime and the presence of subglacial water and upstream basal melting (Franke et al., 2024).





### 5.3 Ice core reconnaissance

One of the fundamental reasons why radar surveys are conducted in Greenland and Antarctica is to identify suitable locations for deep ice core drilling (Mutter and Holschuh, 2025). Moreover, radar measurements near core sites offer a way to extrapolate the information from dated ice cores laterally across a wider area. For this purpose, the European Project for Ice Coring in Antarctica (EPICA) pre-site surveys in western Dronning Maud Land used airborne EMR radar measurements between 1994 and 1999 to search for a region with undisturbed layering, a low flow velocity, and sufficient ice thickness for drilling, ideally

at a summit or ice divide (Steinhage et al., 1999; Steinhage, 2001). Additionally, the bedrock topography played a crucial role in determining a suitable drilling location (Steinhage et al., 1999, 2001).

The ice-thickness data also formed the basis for further ice dynamic modelling leading to the selection of the EPICA DML drilling site near 75°S and 0° (Huybrechts et al., 2000; Wilhelms et al., 2014). Subsequently, Kohnen Station was established in 2001 at the drill site, providing a logistics base for deep ice core drilling of the EDML ice core, which had a final length

of 2774.15 m (Wilhelms et al., 2014). To locate a suitable drill site for the oldest Antarctic ice core, Sutter et al. (2019) used radar-derived ice thickness data to perform 3-D continental ice-sheet modelling. They identified several regions potentially maintaining 1.5 million years old ice around Dome Fuji, Dome C and Ridge B in East Antarctica.

In connection with the North Greenland Ice Core Project (NorthGRIP) for drilling in central Greenland between 1996 and 2001 (Dahl-Jensen et al., 2002), an extensive grid was flown around the drilling site to map internal stratigraphic layering,

which might threaten the continuity of the ice core chronology (Nixdorf and Göktas, 2001), and to connect the new site to the existing GRIP ice core (Dahl-Jensen et al., 1997). The data suggested that, at the time, a continuous ice core record reaching back to the Eemian period (129 – 116 ka BP) in the last interglacial could be expected. Moreover, AWI UWB radar data were used together with NASA's Operation IceBridge data to determine a suitable ice core drill site on Müller Ice Cap on Umingmat Nunaat (Axel Heiberg Island), in the Canadian Arctic (Figure 9c) to retrieve an undisturbed Holocene climate record (Lilien

et al., 2024).

### 5.4 Age-depth distribution of the Antarctic and Greenland Ice sheets

Internal reflection horizons (IRHs) are crucial for understanding the age–depth structure of ice sheets and can often be precisely dated using ice cores. Prominent ice-sheet wide IRHs are typically caused by the deposits of global bipolar volcanic eruptions and can often be traced in radargrams across large areas of Antarctica and Greenland (Millar, 1981). At several ice-core sites

in Greenland (Hempel et al. 2000 at GRIP, Mojtabavi et al. 2022 at NorthGRIP, NEEM and EastGRIP) and Antarctica (Eisen et al. 2006; Franke et al. 2025b at EDML and Winter et al. 2017 at EDC) electromagnetic forward modelling showed that many of the prominent IRHs in Antarctica and Greenland are linked to conductivity peaks in the ice. Consequently, IRHs are present and intercomparable in various radar data products available to the radioglaciology community (Winter et al., 2017; Franke et al., 2025b). Moreover, IRHs represent consistent age markers, serving as archives of the historical development of

the Antarctic and Greenland ice sheets. They are essential for reconstructing the mass balance at both the ice surface and base,




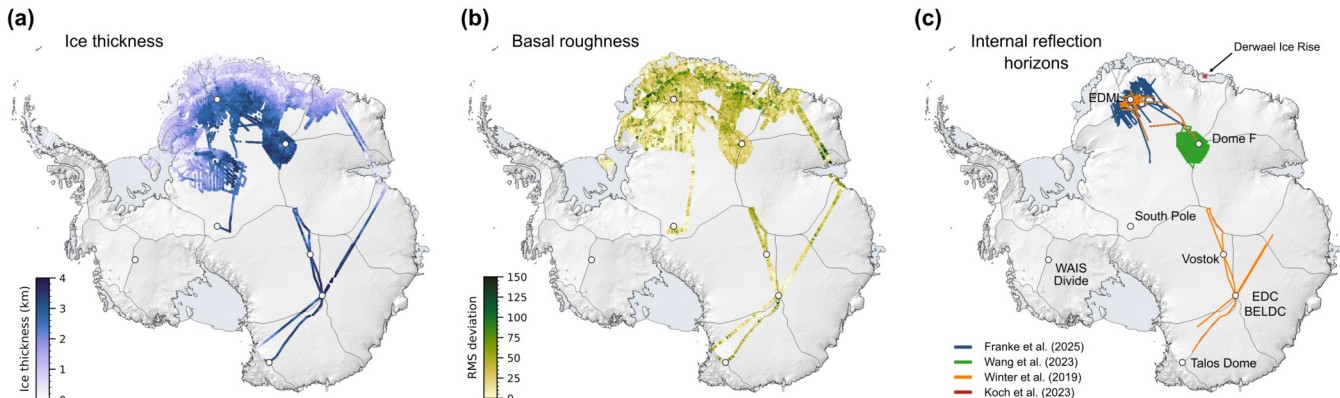

**Figure 11.** Selection of results from AWI radar data over Antarctica: (a) Ice thickness (Steinhage et al., 2001; Frémand et al., 2023; Pritchard et al., 2025), (b) Basal roughness represented as RMS deviation (Eisen et al., 2020), (c) Internal Reflection Horizon (IRH) coverage from Franke et al. 2025b (blue), Wang et al. 2023 (green), Winter et al. 2019 (orange), and Koch et al. 2023 (red).

as well as for understanding the deformation history of the ice, which is currently a major objective of the SCAR Action Group AntArchitecture (Bingham et al., 2025).

IRHs have been extensively traced and dated in AWI radar data across Dronning Maud Land and central East Antarctica (Steinhage et al., 2013; Winter et al., 2019; Wang et al., 2023; Franke et al., 2025b, Fig. 11 c). As part of the fourth International

Polar Year AWI conducted the first Dome Connection flights to connect major ice domes in East Antarctica in 2007–2008 (Talos Dome–Dome Concordia–Vostok–Dome A; Winter et al., 2019). Moreover, IRHs were traced over coastal regions in eastern DML to investigate the dynamic imprints and histories of ice rises (e.g., Halfarryggen and Derwael Ice Rise; Drews et al., 2013; Koch et al., 2023; Ershadi et al., 2024; Henry et al., 2025b).

It has been demonstrated that some of the IRHs traced over large areas in radar data from Dronning Maud Land (Winter

et al., 2019; Wang et al., 2023; Franke et al., 2025b) and Central East Antarctica (Winter et al., 2019) match to similar IRHs in both East and West Antarctica (Franke et al., 2025b). This represents a significant contribution to Antarctic-wide modelling approaches that use IRHs to calibrate their models (e.g., Sutter et al., 2021; Theofilopoulos and Born, 2023) and to develop a three-dimensional, continent-wide age-depth architecture model (as for the Greenland Ice Sheet MacGregor et al., 2015, 2025), as pursued by the AntArchitecture initiative (Bingham et al., 2025).

Furthermore, AWI data have enabled IRHs to be extensively mapped in central and northeast Greenland (Nixdorf and Göktas, 2001; Franke et al., 2023a; Jansen et al., 2024, Fig. 13 b) around the NorthGRIP ice core site and at the onset of the Northeast Greenland Ice Stream. When IRHs are mapped in closely-spaced survey grids, they can be used for constructing 3D surfaces of ice sheet structures (Bons et al., 2016; Franke et al., 2022a, 2023a; Jansen et al., 2024) providing crucial information on ice deformation.





Only a fraction of visible IRHs have been traced to date. The large archive of radar data over the polar ice sheets thus represents a considerable untapped opportunity for tracing IRHs using semi-automatic tracing methods as well as with the help of machine learning (Moqadam and Eisen, 2025; Moqadam et al., 2025).

## 5.5    Ice-sheet dynamics and stability

Besides exploring areas of the ice sheet that are dynamically stable with undisturbed internal stratigraphy, various AWI radar
campaigns have focused on dynamic regions in Greenland and Antarctica, particularly with the aim to better understand ice-stream systems. From 2016 through 2022, surveys using the AWI UWB radar primarily targeted the onset region of NEGIS in the vicinity of the EastGRIP ice core drill site and the region of the 79° NG. The high-resolution radar data collected along and across the ice flow direction at NEGIS, as well as over its shear zones (Franke et al., 2022c), provided fundamental insights into the formation and temporal evolution of this ice stream (Jansen et al., 2024). The data help underscore the implausibility
of the hypothesis linking the fast-flowing NEGIS to exceptional geothermal heat flux (Fahnestock et al., 2001; Bons et al., 2021). Furthermore the data contributed towards a better understanding of the evolution of large scale folding (Zhang et al., 2024; Bons et al., 2025) with implications for ice stream dynamics in northeast Greenland during the Holocene (Franke et al., 2022a). Additionally, the AWI UWB operated in 2018 and 2022 near EastGRIP in swath mode, enabling a high-resolution reconstruction of bedrock topography at the NEGIS onset (Fig. 10; Carter et al., 2025).

Callens et al. (2014) investigated the basal conditions of West Ragnhild Glacier in DML to characterize its ice flow regime using satellite remote sensing, AWI airborne radar data and ice sheet modelling. In coastal areas in eastern DML, AWI radar data collected over ice rises (Matsuoka et al., 2015; Koch et al., 2023) were used as ice dynamic proxies by interpreting the isochrone geometry beneath the local ice divides. Here, stratigraphic arches, referred to as Raymond Bumps (Raymond, 1983) are a metric for the stability of conditions at the ice-divide, which, in turn, is a proxy for the millenial timescale stability of
the surrounding catchment. Examples of surveyed ice rises in Dronning Maud include Halvfarryggen (Drews et al., 2013), Derwael Ice Rise (Koch et al., 2023; Henry et al., 2025a, b), and Hammarryggen ice promontory (Ershadi et al., 2024).

Ice shelves, their interactions with the ocean, and their capacity to buttress tributary ice flows have also been investigated with radar data. This includes, for example, the role of smaller ice rumples which increase ice-shelf damage (Humbert and Steinhage, 2011), the use of highly resolved ice-shelf thickness maps to determine ocean-induced melt rates (Neckel et al.,
2012), partitioning of the ice-shelf composition (Višnjević et al., 2022, 2025), and the imaging of subglacial water outlets near the grounding zone (Drews et al., 2017; Neckel et al., 2021; Zhou et al., 2025).

## 5.6    Ice-sheet and ice-shelf hydrology

Airborne radar data from AWI surveys have played a central role in advancing our understanding of ice sheet hydrology, from subglacial lakes beneath the Antarctic ice sheet (Livingstone et al., 2022; Goeller et al., 2016; Humbert et al., 2018) to englacial
channels (Humbert et al., 2025) and surface melt features in Greenland. Since the early identification of bright, flat reflectors linked to subglacial lakes (Oswald and Robin, 1973), AWI radar systems have been instrumental in detecting both active and stable water bodies beneath ice sheets (Goeller et al., 2016; Humbert et al., 2018; Neckel et al., 2021). While surface elevation





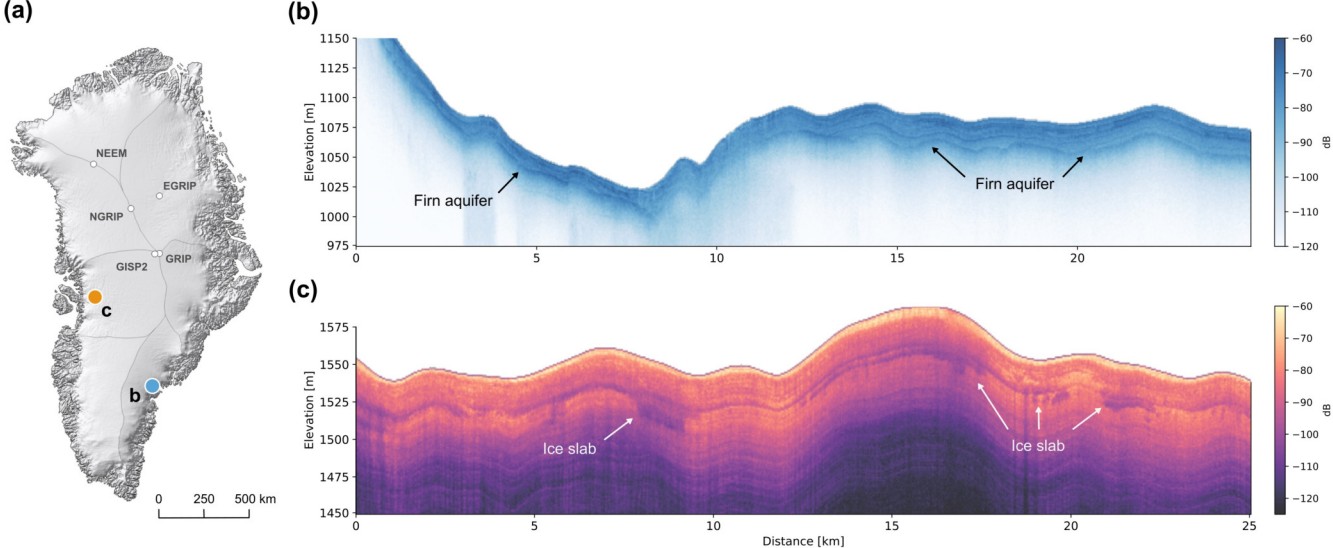

**Figure 12.** RES signatures of (b) englacial meltwater in firn (Profile ID: 20240821_01_010) and (c) refrozen ice slabs (Profile ID: 20240828_01_006). Both radar profiles were recorded with the UWB in a frequency range of 150 − 520 MHz during the ARK 2024 Greenland campaign. The locations of both radar profiles are indicated in (a).

changes help identify actively filling or draining lakes (Neckel et al., 2021), stable subglacial lakes require radar confirmation using criteria such as high basal reflectivity, relative power contrasts, and echo specularity (Humbert et al., 2018). However,

interpretation using these criteria is complicated by uncertain attenuation rates and the presence of basal ice with entrained water (Hills et al., 2024). Additional indicators, such as disrupted internal layering or slope changes in radar horizons, can further support subglacial lake identification (Gudlaugsson et al., 2016; Humbert et al., 2018).

     In Greenland, AWI radar campaigns have contributed key insights into surface meltwater refreezing and firn saturation (Fig. 12). Increasing surface melt has led to the widespread formation of ice slabs—dense, refrozen layers within the firn that

reduce permeability and enhance runoff (MacFerrin et al., 2019). During the 2024 Polarmonitor campaign, AWI used UWB radar in high resolution mode to image stratigraphic detail in these slabs (Fig. 12c). Tracking their expansion is critical for understanding changes in Greenland's water retention capacity and for benchmarking firn hydrology models. Moreover, firn aquifers act as long-term reservoirs of liquid water in southeastern Greenland's accumulation zone and were mapped using AWI radar data as areas showing strong firn-ice reflectors (Fig. 12b) and an absence of bed echoes due to signal attenuation

due to the presence of water (Miège et al., 2016). Transects over regions like Køge Bugt demonstrated that detecting changes in aquifer extent is only possible through airborne radar, as these features are invisible in optical or surface elevation data.

     In Antarctica, AWI radar data contributed to assess potential subglacial lake locations underneath the East Antarctic ice sheet (Goeller et al., 2016). Moreover, radar data in combination with remote sensing products and thermal modelling enabled the detection of (i) cascading subglacial water flow via filling and draining of subglacial lakes (Neckel et al., 2021) as well as (ii)

freeze-on of subglacial water with entrained sediment (Franke et al., 2023b, 2024) at the onset of the Jutulstraumen Glacier.



AWI radar data also revealed the persistence of englacial channels (e.g. Zhou et al., 2025, near the grounding line (GL) of the Roi Baudouin Ice Shelf), as well as channels formed during supraglacial lake drainage at Greenland's 79° NG (Humbert et al., 2025). Here, a 21 km$^2$ lake has repeatedly filled and drained over the past decade and repeated UWB surveys since 2016 revealed englacial features that remain detectable and mobile years after formation (Humbert et al., 2025). These observations offer rare insight into how long such drainage pathways remain open and how they interact with glacier motion. Additional radar observations have shown that lake ice in supraglacial lakes causes a pronounced thickness gradient, forming continuously at the upstream end as the glacier flows downstream (Schröder et al., 2020). This internal structure influences lake stability and drainage behavior, with implications for ice flow.

In addition, fractures and crevasses are important for both hydrology and ice mechanics and are well resolved in AWI radar data. Crack tips appear as hyperbolas in radargrams and can be analyzed to assess crevasse depth and geometry. Radar surveys have detected basal crevasses on the Ross Ice Shelf (Jezek et al., 1979) and surface fractures in Greenland (Forster et al., 2014; Thompson et al., 2020). These measurements have been used to estimate energy release and to identify fatigue cracks (Humbert et al., 2023a, b). Moreover, fractures linked to supraglacial lake drainage are clearly visible in radargrams, offering a new window into meltwater-induced fracturing processes (Humbert et al., 2025).

## 5.7 Mapping and understanding crustal and geological variability and evolution

AWI airborne radar data play an important role in a number of studies of large-scale geological variability and tectonic structure in Antarctica (e.g., Ruppel et al., 2018). Once adjusted for the loading effect of the ice sheet, which requires detailed analysis of coincident gravity anomaly data, the regional height of the upper rock surface can be related to variations in the thickness of the crust. In this way, Riedel et al. (2012) calculated considerable variability about a relatively thick mean crustal thickness in Dronning Maud Land, relating it to the amalgamation and later breakup of the supercontinent Gondwana by tectonic collisions and extension. They noted also that very thick ($\sim 51$ km) crust of the Wohlthat Massif appears to be subject to ongoing vertical isostatic motions in response to much more recent changes in the thickness of the East Antarctic ice sheet.

Radar-derived bed heights are essential for understanding and interpreting those components of the gravity anomaly signal that vary along with lithological variability, especially for calculating the Bouguer anomaly for certain areas. These calculations remove the gravity effects of density contrasts between rocks, ice, and air, showing the regional crustal thickness variability. Using this kind of approach, Riedel et al. (2012) identified long linear density contrasts that mark crustal suture zones: the lines of long-vanished oceans that were destroyed during the incorporation of what is now Antarctica into Gondwana, as well the lower density fill of Jurassic-aged extensional basins close to the coast.

Eagles et al. (2018) and Franke et al. (2021a) used radar-derived depths to the glacial bed to map networks of V-shaped valleys that they interpreted as evidence for the presence of ancient preserved fluvial landscapes in the deep interior of DML. Eagles et al. (2018) noted that some of the valleys appear to have been cut by erosion that was focused onto ancient tectonic structures. They considered the geological system's various roles in the history of sediment routing to the floor of the deep Southern Ocean, and in the long-term topographic evolution of Dronning Maud Land, in particular its seaward-facing great escarpment.





Radar data also play an important role in the interpretation of magnetic field anomaly variability (Mieth et al., 2014; Ruppel et al., 2018).The depth of the magnetic sources can be roughly estimated on its wavelength spectrum or more simply by comparative lineament analyses. Sources that lie deeper than the radar bed depth imply the presence of intervening layers of sedimentary rocks, whose magnetic susceptibilities tend to be small. In this way, Paxman et al. (2019) reported on the presence of a major, probably Jurassic-aged, sedimentary basin between the South Pole and Pensacola mountains. Nogi et al. (2013)

and Guy et al. (2024) elaborated on the structure of the crustal suture zones around Lützow-Holm Bay and further south in DML, some of which were recognized earlier by Riedel et al. (2012) and Ruppel et al. (2018). Here, Guy et al. (2024) conclude that it comprises fragments both of billion year-old volcanic island arcs and at least one much older continental fragment. The remnants of the ancient volcanic system today cover an area $\sim 600\,\text{km}$ wide and nearly $1500\,\text{km}$ long, meaning that whilst active it may have been a feature comparable to today's Antarctic Peninsula or Japan.

## 5.8 Ice shelf bathymetry and stability

The use of AWI airborne radar data have been crucial in enhancing our understanding of subglacial and bathymetric features beneath Antarctic ice shelves. Across multiple studies, these data have been employed to infer bathymetric models, assess ice shelf stability, and explore the dynamics of ice–ocean interactions.

Lambrecht et al. (1999) used AWI EMR radar data from the ANT 1994/95 season to investigate the mass-balance conditions

in the southeastern Ronne Ice Shelf (RIS). Moreover, the radar data contributed to a comprehensive data set of ice thickness of the entire Filchner–Ronne Ice Shelf, with specific focus on grounding lines (e.g., of Foundation Ice Stream, Lambrecht et al. 1997) and marine ice (Lambrecht et al., 2007).

Eisermann et al. (2020, 2021) utilized radar data in coastal Dronning Maud Land to model subglacial topography beneath the Ekström, Atka, Jelbart, Fimbul, Vigrid, Borchgrevink and Roi Baudouin ice shelves. By integrating ice-shelf thickness derived

from radar data with seismic, gravity, and multibeam bathymetric references, the studies revealed key topographic features such as deep troughs and sills that regulate the entry of Warm Deep Water into the cavities beneath ice shelves. These findings underscore the role of bathymetric features in modulating basal melting and shielding even small ice shelves from oceanic heat. Variations in warm water access between different ice shelves highlights the spatial variability of ice–ocean interactions, thereby providing insights into potential impacts of climate-driven changes in water temperature and circulation. Moreover,

the refinement of the regional bathymetry over the Nivl and Lazarev ice shelves emphasizes the protective role of bathymetric ridges (Eisermann et al., 2024) and are important constraints for glaciological and oceanographic models.

These data also helped to determine subglacial water flow at the grounding line, estimating basal melting rates, and assess ice shelf morphology and stability (Drews et al., 2017, 2020; Višnjević et al., 2022, 2025; Henry et al., 2025a, b). On the Fimbul and Jelbart Ice Shelf, AWI radar data helped to constrain the dynamic ice shelf evolution based on their surface and

basal structure (Humbert and Steinhage, 2011; Humbert et al., 2015). Moreover, radar-derived ice shelf and ice rise thickness measurements were key to determine the basal mass balance of Ekströmisen (Neckel et al., 2012) and the evolution of its drainage basin over the last $40\,000$ years (Schannwell et al., 2020).



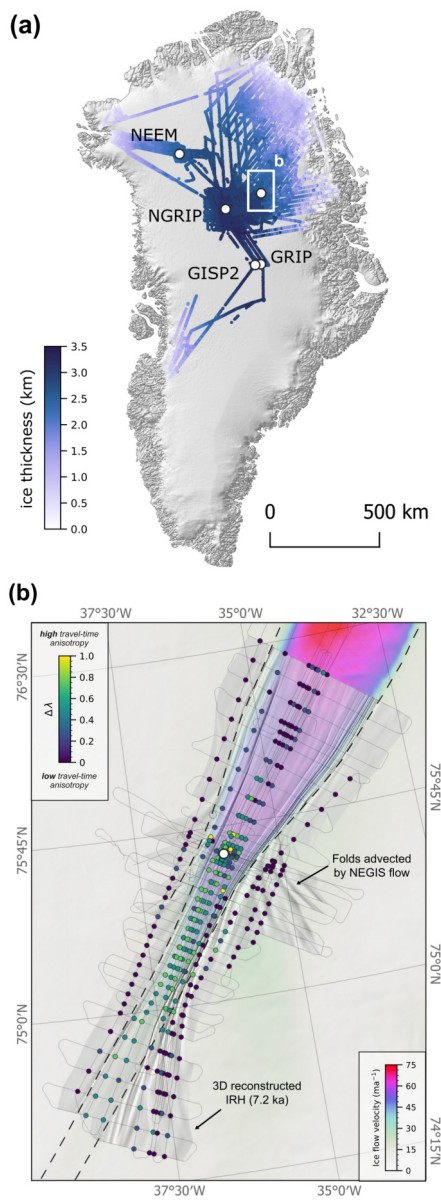

**Figure 13.** Selection of results from AWI radar data over Greenland: (a) Ice thickness. (b) Zoom on the onset region of the Northeast Greenland Ice Stream (NEGIS) centred at the location of the EastGRIP ice core. The background is ice flow velocity (Joughin et al., 2018) overlain by a hillshaded version of the 7.2 ka IRH depth from Jansen et al. (2024). The colored dots represent the depth-averaged difference in horizontal eigenvalues ($\Delta\lambda$) of the ice crystal orientation fabric derived from an analysis of travel-time differences of crossing radar profiles (Gerber et al., 2023). The fine black lines represent the radar profiles of the EGRIP-NOR-2018 survey (Franke et al., 2022c), and the approximate location of the shear margins is indicated with a dashed line.





## 5.9 Mass balance

AWI's radar systems were used to analyze surface mass balance (SMB) and snow accumulation rates in different glaciolog-
ical contexts. Accumulation distribution was mapped on Pine Island Glacier in West Antarctica with the ASIRAS system,
demonstrating identification annual resolution from IRHs on a regional scale (Kowalewski et al., 2021). The integration of
these radar-derived layers with ground-based neutron probe measurements facilitated precise dating and density profiling, cov-
ering over 2 300 km, eventually resulting in SMB. The results highlighted SMB distribution trends, such as the pronounced
orographic precipitation shadow effects in specific regions, and provided critical inputs for modelling total mass input to the
basin without evidence of a long-term trend in accumulation rates (Kowalewski et al., 2021).

Similarly, in Greenland, AWI's use of the ASIRAS provided high-resolution measurements of winter snow accumulation
rates in the percolation zone (Helm et al., 2007). By detecting subsurface reflection horizons corresponding to previous summer
melt surfaces, ASIRAS captured spatial variability influenced by topographic undulations and snow redistribution by winds.
This method offered accumulation estimates comparable to field measurements and demonstrated the potential for extended
regional and temporal monitoring of accumulation rates (Helm et al., 2007) similar to the usage of ground-based radar systems
in Greenland (Steinhage et al., 2005; Karlsson et al., 2020).

Most studies that focused on SMB reconstruction in DML used the radar stratigraphy from shallow ground-penetrating
radar surveys (Rotschky et al., 2004; Anschütz et al., 2006, 2008) in combination with data from firn cores. However, with its
capability to resolve near-surface IRHs when operated in wideband mode (Fig. 4e and 5b), the AWI UWB radar supports SMB
reconstructions over wider areas (Zuhr et al., 2025) and in combination with deep-sounding surveys (Fig. 6d).

At the onset of the Jutulstraumen Glacier in western Dronning Maud Land, the UWB radar was used to investigate basal re-
flection units that likely result from the refreezing of basal meltwater originating from further upstream. These units contribute
to a positive contribution of basal mass balance in this area (Franke et al., 2024).

## 5.10 Ice-flow Modelling

AWI radar data have significantly advanced ice sheet modelling by providing detailed insights into the internal structure and
dynamics of ice sheets. A key application has been to identify potential drill sites for retrieving the oldest ice in Antarctica. By
combining radar derived isochrone stratigraphy with 1D to 3D thermo-mechanical ice flow models, Wang et al. (2023) were
able to refine the age-depth scales and basal thermal conditions in the Dome Fuji region. In combination with ground-based
radar data, Chung et al. (2023) revealed spatial variability in basal melt rates and areas with stagnant ice, helping to constrain
models of basal thermal states and accumulation rates (e.g., Karlsson et al., 2018).

Radar-derived IRHs also play a critical role in calibrating ice-sheet models in both ice sheets (Gerber et al., 2021; Sutter
et al., 2021; Višnjević et al., 2022, 2025). IRHs provide 3D observational constraints that highlight discrepancies in surface
mass balance, basal drag parameterizations and flow dynamics. By validating models with observed internal stratigraphy, it
can be demonstrated how localized variations in e.g., geothermal heat flux, subglacial topography, and surface mass balance
affect long-term ice-sheet evolution (Sutter et al., 2021). Moreover, models were used to predict the age stratigraphy of locally





accumulated ice on the Roi Baudoin ice shelf for a given set of oceanic and atmospheric boundary conditions (Višnjević et al., 2022, 2025).

At the onset of NEGIS, Gerber et al. (2021) utilized isochrones traced from AWI radar data to constrain a two-dimensional ice flow model, revealing that upstream flow introduces climatic biases in ice cores due to advection of snow deposited under different conditions. By modelling backward particle trajectories, they determined source locations and past accumulation rates, highlighting the significant roles of basal melting and sliding in NEGIS dynamics. Based on the englacial folds at NEGIS (Jansen et al., 2024), Zhang et al. (2024) modeled the effect of ice anisotropy and density variations on fold amplification into large-scale folds during flow. Moreover, Gerber et al. (2023) used radar data and ice-flow modelling to infer the influence of the mechanical anisotropy of NEGIS (further details in Section 5.11).

AWI's RES-derived ice thickness data alone were widely used for ice sheet modelling efforts (Humbert et al., 2010; Kleiner and Humbert, 2014; Morlighem et al., 2017, 2020; Rückamp et al., 2019a, b, 2020; Schannwell et al., 2019, 2020; Mohammadi-Aragh et al., 2025). While many modelling studies require accurate geometries and hence ice thickness data over larger domains, up to the entire ice sheet, other modelling approaches are benefiting substantially from ice thickness profiles along glacier streamlines. Modelling shows that precise overflight of a streamline is important for the accuracy of the results. Along-flow modelling approaches require bedrock topography data along the precise streamline of the glacier for two reasons: (i) the influence of the roughness and small scale topography on sliding needs to be represented well and/or (ii) the gravitational load has to be realistic. This may not be given in gridded, interpolated datasets from which a transect is extracted. Examples of such studies are Humbert et al. (2015) and Christmann et al. (2021).

AWI radar data have been instrumental in constraining ice material behavior and evaluating glacier dynamics. For both an Antarctic ice shelf and a grounded-floating transition zone (Humbert et al., 2015; Christmann et al., 2021), radar transects aligned with flowlines were combined with surface elevation or in-situ motion data. This setup allowed simulations to test different constitutive relations under controlled conditions, excluding lateral effects.

In model validation efforts, AWI RES-derived ice geometry was used to assess the adequacy of the Blatter-Pattyn higher-order approximation compared to full-Stokes simulations (Rückamp et al., 2022). Results showed that when model resolution is coarser than a certain fraction of ice thickness, complex physics offers no advantage, underscoring the need for dense geometry data. The study also highlighted how reliable conclusions are only possible when ice thickness is well-resolved, as demonstrated by contrasting simulations of two neighboring outlet glaciers, only one of which had dense radar profile coverage.

At Greenland's 79° NG, AWI radar data enabled high-resolution viscoelastic modelling of tidal forcing, basal lubrication, and elastic deformation (Christmann et al., 2021). The study revealed how variations in bed topography influence basal sliding and stress transmission kilometers upstream of the grounding line—demonstrating the critical role of elastic processes and the value of high-resolution RES-derived geometry for capturing short-term glacier dynamics.

## 5.11 Ice crystal fabric

The orientation of crystal lattice axes in ice grains, referred to as the COF, significantly affects both the propagation of electromagnetic waves through ice and the mechanical properties of ice. This anisotropy affects the mechanical properties of ice but





also influences the propagation of radar waves and is, thus, a key focus in the analysis of AWI radar data. In Dronning Maud Land in East Antarctica, radar data combined with the EDML ice core were used to investigate the effect of COF variations on radar wave propagation (Drews et al., 2012). Studies showed that changes in COF, observed at different depths, corresponded to consistent radar reflections (Eisen et al., 2007). These observations also helped explore the origin of the so-called echo-free zone discussing the role of COF (Drews et al., 2009).

In Northeast Greenland at the onset of NEGIS, UWB radar measurements in 2018 (Franke et al., 2022c) were pivotal in improving our understanding of how COF around the EGRIP drill site (Stoll et al., 2025) impacts the ice stream's mechanical anisotropy (Gerber et al., 2023), influencing the directional flow and deformation properties of the ice (Figure 13b). Travel time delays of internal reflection horizons from intersecting radar profiles and the radar beat signatures revealed the effect of COF anisotropy on the ice's hardness during deformation (Gerber et al., 2023). The study further suggests that compared to isotropic

ice, certain parts of the ice stream are significantly harder for along-flow extension and compression, while the shear margins could be softened for horizontal-shear deformation by a factor of two. Subsequent analysis of the characteristic wavelength of folds in cloudy bands of the EGRIP ice core in the cm-scale and folds in radargrams in the km-scale confirms the strong mechanical anisotropy of ice due to the lattice preferred orientation at NEGIS (Bons et al., 2025).

## 6  Data quality

The data quality of AWI's airborne radar systems reflects both the technological evolution of the instruments and the environmental challenges inherent in surveys over polar ice sheets. Over the past three decades, AWI has maintained a high standard of data integrity across its six radar systems, though variations exist due to system-specific characteristics, hardware improvements, and operational conditions. Generally, the datasets are robust and well-suited for scientific analysis, with only isolated profiles occasionally affected by technical issues or electromagnetic noise.

The EMR system, which has undergone continuous development since 1994, exhibits some variability in data quality. For example, defects in hardware components led to a degradation in data quality for the EMR system, particularly during the ANT 2012/13 season and partly during the ANT 2013/14 season. In occasional cases, EMR profiles may contain a mix of results from the short-pulse and long-pulse product.

The ACCU and SNOW systems occasionally show horizontal stripes or aliasing effects, particularly impacting the visibility

of internal reflection horizons in regions with low accumulation rates. In several seasons, particularly from 2015 onward, persistent horizontal stripes affect parts of the radargrams in ACCU data. Additionally, radargrams occasionally exhibit aliasing effects when signals fall outside the range gate, resulting in horizontally flipped sections.

The UWB and UWBM systems, representing the latest generation of AWI's radar systems, consistently deliver the highest data quality, with exceptional resolution and minimal noise. The UWB system, in particular, has demonstrated reliable

performance in both deep and shallow sounding modes, enabling detailed imaging of englacial and subglacial features. Only some profiles from the ANT 2016/17 season in the 24-channel configuration and ARK 2016 season were acquired for testing purposes and may be of reduced quality. The UWBM system, while limited to a smaller number of flights over grounded ice,





provides ultra-high-resolution data, though it is occasionally affected by coherent noise due to its GHz frequency range. The ASIRAS system, primarily used for near-surface studies, generally offers excellent data quality, though some profiles may

suffer from poor signal-to-noise ratios or coherent noise interference.

## 7  Data access and usage

The AWI radar data are publicly available and can be accessed as follows: (1) The inventory of the radar data archive and the location of radar profiles in Antarctica and the Arctic can be viewed via the *Radar Data Viewer over Polar Ice Sheets* in the Marine Data Portal (https://marine-data.de/viewers/). (2) The radar data files are archived and accessible via PANGAEA

(Felden et al., 2023; Eisen et al., 2024, https://doi.org/10.1594/PANGAEA.972094). Radar data from future campaigns will be continuously added to both the Radar Data Viewer and PANGAEA.

### 7.1  Marine Data Portal (Viewer and Metadata)

#### 7.1.1  General Features and Capabilities

AWI's radar data are visualized in the Radar Data over Polar Ice Sheets Viewer of the Marine Data Portal (https://marine-data.

de/viewers) through an interactive web map that allows users to explore and access the data. The Radar Data Viewer is modeled after existing online portals that visualize radar data in polar regions, such as the Polar Airborne Geophysics Data Portal (https://www.bas.ac.uk/project/nagdp/) of the British Antarctic Survey (BAS; Frémand et al., 2022) or the Open Polar Server GeoPortal (https://ops.cresis.ku.edu/) of the Center for Remote Sensing of Ice Sheets (CReSIS) at the University of Kansas. The Marine Data Portal is a brand of the Earth Data Portal (Heß et al., 2023) with a focus on content from the marine community

and is coordinated by the German Marine Research Alliance (Deutsche Allianz Meeresforschung). Using an interoperable Web Map Service (WMS) hosted by the Observations to Archives and Analysis (O2A) Spatial Data Infrastructure (SDI) (Konopatzky et al., 2023), the map displays the locations of AWI radar profiles, categorized by radar system, and includes profiles from CReSIS and BAS to provide context with data from other institutes and different background maps. The external profiles from BAS and CReSIS are not available for download but help to create a comprehensive overview of radar data in the

Arctic and Antarctic.

#### 7.1.2  Radar Profiles, Metadata and Radargram Visualization

Users can filter displayed profiles by data acquisition period and radar system using the left-side panel (Fig. A3). The bottom panel enables toggling between different base layers, and a globe icon allows switching between a global perspective and polar stereographic projections for the Arctic (EPSG 3995) and Antarctica (EPSG 3031). Selecting a radar profile segment opens a

popup window with selected metadata such as the radar system, campaign name, principal investigator (PI), project name, and the profile ID. The "show more" option in the popup opens a side panel on the right (Fig. A4), which provides detailed metadata,





a quicklook radargram of the selected radar profile and DOI link for downloading the dataset from PANGAEA. Quicklooks can be enlarged by selecting them, and all quicklooks from a radar campaign can be viewed using the "View Gallery" option.

For UWB and UWBM systems, quicklooks are shown in decibels (dB), while EMR, ACCU, SNOW, and ASIRAS systems are displayed in automatic gain control (AGC) mode. EMR, ACCU and SNOW profiles can be several hundreds of kilometers long. The quicklooks in the radar viewer are thus subdivided into 100 km-long segments. The data for download are however provided in the full profile length. Picks for the surface and, where available, base reflections are included. If a profile selection on the map is ambiguous, multiple matching profiles are shown, which users can browse using the "next" and "previous" buttons.

### 7.1.3 Background Maps and Auxiliary Data

To place the radar data in a glaciological context, we provide background maps of subglacial bed topography and ice surface flow velocities for both Antarctica and Greenland. In the Radar Viewer, the Antarctic background layers currently include the Bedmap3 bed topography (Pritchard et al., 2025) and MEaSUREs ice flow velocity data (Rignot et al., 2017). Additionally, the Landsat Image Mosaic of Antarctica (LIMA; Bindschadler et al., 2008) can be selected as a background map. For Greenland, users can choose the BedMachine bed topography (Morlighem et al., 2017) and the MEaSUREs Multi-year Greenland Ice Sheet Velocity Mosaic (Joughin et al., 2018) as background maps.

### 7.2 PANGAEA (Archive and Download)

The AWI radar data products are published in PANGAEA and are directly available for download. The data are organized by campaign or season as consolidated entries, with datasets linked to official AWI campaign and flight numbers (events), sensor types, and general geographic locations. The dataset titles include fixed terms indicating the Arctic or Antarctic season (e.g., ARK 1998 or ANT 2018/19), the radar system used (e.g., radar data for the EMR system, ultra-wideband (microwave) radar data for the UWB/M systems, accumulation radar data for the ACCU system, and snow radar data for the SNOW system), as well as the survey region. Searchable keywords cover specific radar systems (e.g., AWI EMR, AWI UWB, AWI UWBM, AWI ACCU, AWI SNOW, ASIRAS), geographic locations, and ice core sites if profiles are near an ice core.

The DOIs for the datasets can be found via the Marine Data Portal, through the "Collection of datasets from AWI's radar systems on ice sheets and glaciers" (Eisen et al., 2024), or by using relevant keywords in PANGAEA's search interface (e.g., https://https://www.pangaea.de/ to search for "AWI EMR" related datasets). Data downloads are facilitated through the "View dataset as HTML" option, which allows users to download individual files or the complete dataset as a compressed *ZIP* archive or an uncompressed *TAR* archive (Fig. A5). Related data publications, such as those on ice thickness, bed topography, internal reflection horizons, and basal roughness, are also listed. All data sets and also complementary data sets from expeditions with AWI's polar aircraft and are available via PANGAEA's Expedition Portal (https://www.pangaea.de/expeditions/ - see links below "Aircrafts" for a comprehesive campaign list and links to campaign data).

The radar products are available in netCDF format. In addition to the primary data matrix, supplementary information is included such as the two-way travel time (TWT) to the ice surface and, where available, the ice base reflection—along with





metadata like sensor height above ground, along-track distance, and more. KML files for radar profile locations and quicklook images are also provided.

For the EMR, ACCU, SNOW, and ASIRAS systems, one data product is provided. For the UWB and UWBM systems, multiple products are available. UWB data include an unfocused (qlook) and a SAR-focused (standard) product. If the UWB system operated in polarimetric mode (e.g., during the ARK 2022 season at the NEGIS onset), data for all four polarizations (VV, VH, HH, HV) are provided. Similarly, the UWBM datasets include products for all four polarizations.

### 7.3 Data usage requirements

AWI radar data can be freely used, but only for scientific purposes (CC-BY-NC). Any other use must be coordinated with the radar data administrators. A condition for publishing with AWI radar data is to cite the corresponding dataset on PANGAEA, this paper, and, if available, a relevant original scientific publication. All publications associated with the dataset are linked on the individual PANGAEA datasets. For datasets that do not yet have a PANGAEA entry (e.g., those that are relatively new or under embargo), but are visible in the Radar Viewer, the overarching PANGAEA entry for the collection is referenced (Eisen et al., 2024). Here, and for all other issues, the Principal Investigators of the respective campaigns should be contacted directly.

## 8 Future directions and extending the impact of AWI's radar data archive

AWI's radar data viewer provides access to a live data archive that is continuously supplemented and expanded. In future steps, the addition of important high-level products derived from the radar data is planned — for example, revised bed reflections and ice thicknesses. Furthermore, the integration of additional data from other acquisition systems collected during survey flights is being pursued, such as airborne laser scanner data to derive surface topography and reflectivity, as well as orthophotos from various nadir-looking optical camera systems like the Modular Airborne Camera System (Neckel et al., 2023). These will provide valuable insights into snow and ice surface characteristics, which help in better analyzing radar reflections. Examples include surface crevasses, supraglacial lakes, and topographic surface roughness.

## 9 Conclusions

Over the past 30 years, since 1994, AWI has conducted more than 40 seasons of airborne radar campaigns in Antarctica and Greenland collecting more than 1 610 000 profile-km of radar data. Six different radar systems have been utilized with capabilities meeting a range of demands in the fields of ice thickness sounding and high-resolution imaging of englacial layers. The AWI radar systems cover a wide range of penetration depths, reaching down to 4 km with the EMR and UWB systems (with range resolutions between 0.35 – 50 m), and high-resolution shallow-sounding radars, such as the Accumulation Radar (ACCU), Snow Radar (SNOW), and UWBM, with provide range resolutions between 1 – 50 cm.

In Antarctica, the radar surveys have primarily focused on East Antarctica's Dronning Maud Land and the central East Antarctic region. In Greenland, AWI radar surveys predominantly cover the northern and northeastern areas, with numerous





surveys also conducted in southwest Greenland. The platforms used to operate the radar systems are AWI's Polar aircraft Polar 2, 5, and 6. These aircraft have been equipped with wheels and skis, allowing them to operate flexibly on ice sheets.

The initial scientific objectives in Antarctica and Greenland primarily focused on determining ice thickness and conducting reconnaissance surveys for the EDML drilling site (Antarctica) and e.g., NorthGRIP (Greenland). At present, however, research questions now span all subdiciplines of glaciology and polar geology, including ice thickness determination, basal properties

of ice sheets, internal stratigraphic architecture, ice dynamics, crustal properties and their evolution, ice shelf characteristics and stability, as well as the underlying bathymetry, surface and basal mass balance, ice-sheet modelling, and the analysis of ice crystal fabric and its effects on ice flow.

The AWI radar data are publicly available and adhere to a common data standard, simplifying public access, use, and analysis. Through the Marine Data Portal, all radar surveys can be viewed on an interactive online map with quicklook radargrams,

and the data can be downloaded from the PANGAEA Data Publisher. The radar data are provided in a FAIR-compliant data format (netCDF files) along with the profile locations and include relevant supplementary information, such as ice surface reflection, flight altitude, and bed reflection (if available).





**Appendix A:  AWI radar survey seasons in Antarctica and Greenland**

AWI radar data in Antarctica primarily cover Dronning Maud Land, parts of central East Antarctica, and small sections of West
Antarctica (Figure 8). Detailed information about individual campaigns for each Antarctic season, including their coverage,
campaign names, survey regions, platforms, radar systems used, and flown survey kilometers, is provided in Figure A1 and
Table A1.

In general, the data show that the EMR system was exclusively used until 2011. Between 2012 and 2017, this was supple-
mented by flights using the Accumulation Radar (ACCU) and Snow Radar (SNOW). Since 2018, surveys have predominantly
been conducted with newer systems such as UWB and UWBM. It is important to note that the ASIRAS system is not included
in the campaign-wise overview, as it was mostly used in collaborations with other institutes. Only in a few exceptional cases
was the ASIRAS system mounted on one of AWI's Polar aircraft.





**Figure A1.** Season-wise maps of AWI radar sounding campaigns in Antarctica with the different radar systems (EMR = red, ACCU = green, SNOW = cyan, UWB = dark blue, UWBM = purple). Note that the profile lines of the ASIRAS system are not represented here. The white dots represent deep ice core sites and the black lines the IMBIE drainage basins (Rignot et al., 2019).



**Table A1.** Season-wise overview of airborne radar surveys in Antarctica with the Campaign acronyms, approximate survey region, radar instrument and platform used as well as the respective survey-km flown per instrument.

| Season | Campaigns | Survey Region | Platform | Instrument | Survey-km |
|---|---|---|---|---|---|
| ANT 1994/95 | MAGRAD | South of Halley, Berkner Island and Foundation Ice Stream | Polar 2 | EMR | 18 562 km |
| ANT 1995/96 | EPICA I | Western DML, EPICA pre-site survey | Polar 2 | EMR | 20 749 km |
| ANT 1996/97 | EPICA II | Western DML, EPICA pre-site survey | Polar 2 | EMR | 13 628 km |
| ANT 1997/98 | EPICA III | Western DML, EPICA pre-site survey | Polar 2 | EMR | 18 154 km |
| ANT 1998/99 | EPICA IV | Western DML, EPICA pre-site survey | Polar 2 | EMR | 21 154 km |
| ANT 2000/01 | SEAL, VISA | Western DML, Jutulstraumen, DML GL | Polar 2 | EMR | 15 907 km |
| ANT 2001/02 | SEAL, VISA | Western DML, Jutulstraumen GL, EDML | Polar 2 | EMR | 44 641 km |
| ANT 2002/03 | EPICA, SEAL | Western DML, Ekströmisen, Coats Land, EDML, Dome Fuji | Polar 2 | EMR | 27 381 km |
| ANT 2003/04 | EPICA, SEAL, VISA | Central DML, EDML, Princess Ranghild Coast | Polar 2 | EMR | 41 414 km |
| ANT 2004/05 | VISA | DML | Polar 2 | EMR | 15 053 km |
| ANT 2005/06 | ANTSYO | Ekströmisen, Shireas Glacier basin | Polar 2 | EMR | 17 056 km |
| ANT 2007/08 | DoCo, VISA | Central DML, Vostok, Dome C, Talos Dome | Polar 5 | EMR | 35 963 km |
| ANT 2008/09 | RBI, WEGAS | Eastern DML, Berkner Island | Polar 5 | EMR | 46 503 km |
| ANT 2010/11 | GEA I, WEGAS, SRG | Central East Antarctica | Polar 5 | EMR, ACCU | 3 840 km |
| ANT 2011/12 | GEA II, WEGAS | Western DML, and Sør Rondane Mountaines | Polar 6 | EMR, ACCU | 30 736 km 25 100 km |
| ANT 2012/13 | GEA III, WEGAS, RECISL, MaBaJu 1 | Western DML, Recovery Glacier, Sør Rondane Mountaines | Polar 6 | EMR, ACCU | 35 112 km 8 114 km |
| ANT 2013/14 | AMASIN, WEGAS, SWIT, STRUCGLAC, MABAJU 2, RECISL, Sör-Mond, CoFi Structure, VELMA | DML, Antarctic Peninsula, Jelbartisen | Polar 6 | EMR, ACCU, SNOW | 3 037 km 42 703 km 14 603 km |
| ANT 2014/15 | GEA IV | DML | Polar 6 | EMR, ACCU | 1 408 km 11 733 km |
| ANT 2015/16 | GEA-Va-FMA | Western DML, Fimbul Isen | Polar 5 | EMR | 11 830 km |
| ANT 2016/17 | GEA-Vb-FMA, OIR Dome F | DML, Dome Fuji, Jutulstraumen | Polar 6 | EMR, UWB | 25 355 km 1 401 km 17 074 km |
| ANT 2017/18 | ANIRES | Coastal western DML | Polar 6 | EMR | 1 072 km |
| ANT 2018/19 | JuRaS, CHIRP | Jutulstraumen, and GL at Princess Ranghild Coast | Polar 6 | UWB | 20 657 km |
| ANT 2022/23 | RIISERBATHY (GEA VI), Kottas-Aero | Riiser-Larsen Isen, Kottas Traverse | Polar 5 | EMR, UWBM | 13 937 km 1 897 km |
| ANT 2023/24 | RINGS-DML/EL, CHARISO, DML_SnACC | Western DML (GL and divides) | Polar 6 | UWB | 38 499 km |
| ANT 2024/25 | RINGS-DML/EL, CHARISO, CHIRP 2, SANAS, SISI | DML (ice shelves, GL and divides) | Polar 6 | UWB | 21 720 km |



AWI radar surveys in the Arctic have been almost exclusively focused on Greenland, with one exception in 2023, when flights were conducted over the Canadian Möller Ice Cap (Figure A2 and Table A2). Between 1996 and 2012, surveys primarily

targeted northern central Greenland and were carried out using the EMR and ACCU systems. Moreover, in this publication we exclude survey flights over Arctic sea ice.

Since 2013, surveys have shifted to near-coastal regions, particularly in the northeast, but also in the northwest and southwest. In 2018 and 2022, two extensive surveys concentrated on the onset region of NEGIS around the EastGRIP ice core drill site, using the UWB system. Additionally, in 2018, the UWBM system was deployed upstream of the 79° NG. It is important to

note that the ASIRAS system is not included in the campaign-wise overview, as in the Arctic the system was mainly flown by The Technical University of Denmark (DTU).




**Figure A2.** Season-wise maps of AWI radar sounding campaigns in the Arctic with the different radar systems (EMR = red, ACCU = green, SNOW = cyan, UWB = dark blue, UWBM = purple). Note that the profile lines of the ASIRAS system are not represented here. The white dots represent deep ice core sites and the black lines the IMBIE drainage basins (Mouginot et al., 2017).



**Table A2.** Season-wise overview of airborne radar surveys the Arctic (Greenland and Canada) with the Campaign acronyms, approximate survey region, radar instrument and platform used as well as the respective survey-km flown per instrument.

| Season | Campaigns | Survey Region | Platform | Instrument | Survey-km |
|---|---|---|---|---|---|
| ARK 1995 | Nord GRIP | NorthGRIP | Polar 2 | EMR | 8 776 km |
| ARK 1996 | Nord GRIP | North GRIP deep ice core drill site | Polar 2 | EMR | 14 312 km |
| ARK 1997 | Nord GRIP | North GRIP deep ice core drill site | Polar 2 | EMR | 9 231 km |
| ARK 1998 | NOGRAM | Northeast Greenlandbetween North GRIP and Station Nord | Polar 2 | EMR | 10 436 km |
| ARK 1999 | NOGRAM | Northeast Greenlandbetween North GRIP and Station Nord | Polar 2 | EMR | 14 574 km |
| ARK 2004 | NorthGRIP | NE GrIS North GRIP the ice divides towards Camp Century and GRIP/GISP2 | Polar 2 | EMR | 23 919 km |
| ARK 2010 | NEEM & EGIG | Vicinity of the NEEM deep ice core drill site | Polar 5 | EMR ACCU | 20 748 km 9 994 km |
| ARK 2012 | Structure NGT 2012 | Between Dy3 and South Dome and North Greenland Traverse drill sites | Polar 6 | ACCU | 8 368 km |
| ARK 2013 | Top79.5 2013 | Northeast Grenland (Academy Glacier, Hagen Brae, 79° NG) | Polar 5 | EMR | 813 km |
| ARK 2015 | MABANG1, PRESURV 79 | 79° NG and Zacharias Isbrae | Polar 6 | EMR, ACCU | 10 175 km 5 215 km |
| ARK 2016 | Hiawatha, RESURV 79 | Hiawata region, upstream of 79° NG and SW Greenland | Polar 6 | UWB [1] UWBM | 4 602 km 2 347 km |
| ARK 2018 | EGRIP-NOR-2018, RESURV 79, FINEGIS | upstream of 79° NG and NEGIS | Polar 6 | UWB UWBM | 16 658 km 4 651 km |
| ARK 2021 | 79° NG-EC, HTRES | 79° NG and Hans Tausen Ice Cap | Polar 5 | UWB | 5 533 km |
| ARK 2022 | NEGIS-Flow, NEGIS-ANISO, NEGIS Folds | NEGIS near the East GRIP deep ice core drill site | Polar 5 | UWB [2] | 7 151 km |
| ARK 2023 | Müller Ice Cap | Müller Ice Cap, Axel Heiberg Island. | Polar 5 | UWB | 1 052 km |
| ARK 2024 | SLOGIS, ATIWIK Polarmonitor | Upstream of 79° NG and SW Greenland | Polar 6 | UWB | 9 827 km |

[1] The UWB was flown in a 24-channel configuration this season.

[2] Parts of the ARK 2022 UWB survey at NEGIS onset were flown in a polarimetric mode.







**Figure A3.** Screenshots from the Marine Data Portal Radar Data Viewer. (a) Highlights the initial view and key features in the left side panel. (b) Shows AWI UWB profiles in Antarctica's Dronning Maud Land with ice flow velocity (Rignot et al., 2017) in the background. (c) Shows the same setting as in (b) but with bed topography (Pritchard et al., 2025) in the background.



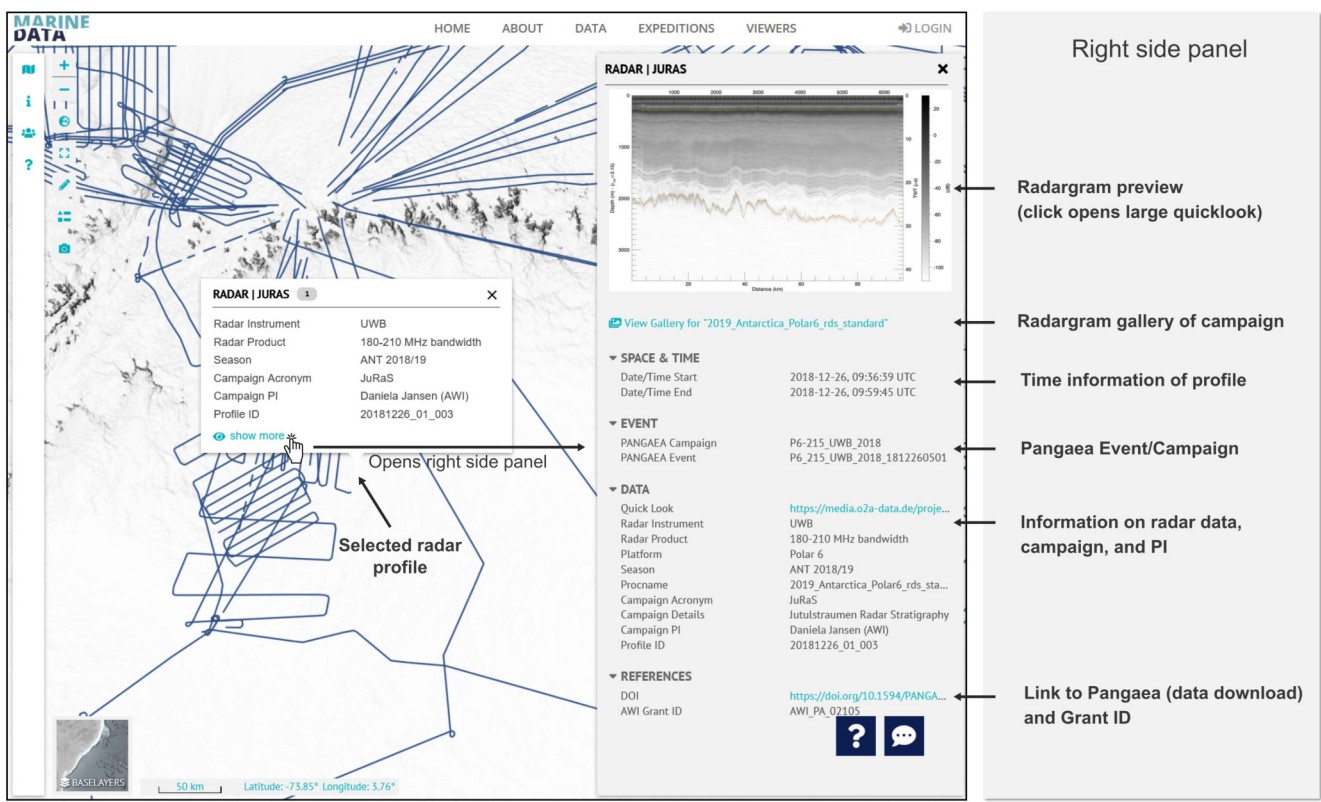

**Figure A4.** Screenshots from the Marine Data Portal Radar Data Viewer highlighting detailed profile information.





**Figure A5.** PANGAEA entry for data download using the example of the ANT 1997/98 EMR survey. Panels (a) to (c) each represent sections of the full Pangaea entry, containing more information than is displayed here. (a) Header section of the PANGAEA entry with a citable reference, abstract, keywords, related publications and Pangaea entries, etc. (b) Data download section, where the "View dataset as HTML" button allows access to the data view. Note that the Data section will only appear if the "View dataset as HTML" button is used. (c) Data presentation with options to download the entire dataset as a compressed ZIP or uncompressed TAR archive, or as individual files. Important items for the data download are highlighted with an orange outline.



*Code and data availability.* The OPR Toolbox (Open Polar Radar, 2023) for radar data processing is available at https://gitlab.com/openpolarradar/opr/. AWI's airborne radar data and metadata can be viewed through the *Radar Data over Polar Ice Sheets* viewer of the Marine Data Portal (https://marine-data.de/viewer/). A comprehensive collection of datasets from AWI's radar systems on ice sheets and glaciers is available in
the PANGAEA dataset bibliography by Eisen et al. (2024): https://doi.org/10.1594/PANGAEA.972094.

AWI's radar data archive is continuously extended and updated with data from new campaigns, improved processing products of existing data, harmonization for better usability, and the integration of ice surface and ice base picks. At the time of this publication, a large portion of the radar campaigns in Antarctica and the Arctic are available for download on PANGAEA. The remaining data are being progressively added. However, all data can also be provided upon request to the corresponding authors of this article. Radar data acquired within the most
recent two years of a request may be subject to an embargo.

*Author contributions.* SF wrote the paper with contributions from DS, VH, OE, AH, and GE. DS, VH, SF, and PHA compiled AWI radar data and performed quality checks. VH provided software for data format conversion. DS, VH, UN, OE, SF, DJ, TB processed AWI radar data. TB implemented the AWI UWB system with support from DJ. AD provided support for the data integration into the PANGAEA Data Publisher. AW, PK, RH, AH, RK, and VH implemented the Radar Data Viewer in the Marine Data Portal. OE coordinated the implementation
of the Radar Data Viewer with support from SF, VH and DS.

*Competing interests.* The authors have the following competing interests: At least one of the (co-)authors is a member of the editorial board of The Cryosphere.

*Acknowledgements.* Logistical support in the field in Antarctica over the past three decades has been provided by the three Neumayer Stations (Germany), Troll Station (Norway), Filchner Station Kohnen Station (Germany), Princess Elisabeth Station (Belgium), Novolazarevskaya
Airbase (Russia), SANAE-IV Station (South Africa), Halley and Rothera Station (Great Britain), Syowa Station and S17 (Japan), Vostok Station Russia, Concordia Station (France & Italy), Mario Zucchelli Station (Italy), and Zhongshan Station, (China). In Greenland we received logical support by Station Nord (Arctic Command, Denmark) and the deep ice drill sites NGRIP, NEEM, and EGRIP (Denmark). We thank the entire AWI hangar team, their technician support in the field as well as support from Aerodata, Optimare, and Fielax during airborne radar campaigns. In addition, we thank the AWI logistics team for enabling airborne radar surveys on both hemispheres. We
acknowledge financial and logistical support from the German Federal Institute for Geosciences and Natural Resources (BGR) to conduct some of the surveys. We thank John Paden and the CReSIS team for long-term support with AWI's airborne UWB and UWBM system. In addition, we thank Mathieu Morlighem, Joe MacGregor and Julien Bodart for providing radar survey line coverage data for Greenland and Antarctica. Moreover, we thank Charlotte Carter, Alexandra Zuhr, and Hameed Moqadam for contributing checking and fixing ice surface picks.
We acknowledge the use of software from Open Polar Radar generated with support from the University of Kansas, NASA grants 80NSSC20K1242 and 80NSSC21K0753, and NSF grants OPP-2027615, OPP-2019719, OPP-1739003, IIS-1838230, RISE-2126503, RISE-2127606, and RISE-2126468. The authors would like to thank Aspen Technology, Inc. for providing software licenses and support. We



acknowledge support via airborne radar campaign funding grants AWI_PA_02001, AWI_PA_02002, AWI_PA_02003, AWI_PA_02009,
AWI_PA_02010, AWI_PA_02011, AWI_PA_02017, AWI_PA_02020, AWI_PA_02027, AWI_PA_02028, AWI_PA_02030, AWI_PA_02031,
AWI_PA_02034, AWI_PA_02037, AWI_PA_02039, AWI_PA_02040, AWI_PA_02041, AWI_PA_02043, AWI_PA_02047, AWI_PA_02050,
AWI_PA_02051, AWI_PA_02054, AWI_PA_02055, AWI_PA_02056, AWI_PA_02058, AWI_PA_02066, AWI_PA_02068, AWI_PA_02072,
AWI_PA_02073, AWI_PA_02077, AWI_PA_02078, AWI_PA_02079, AWI_PA_02080, AWI_PA_02083, AWI_PA_02084, AWI_PA_02085,
AWI_PA_02090, AWI_PA_02092, AWI_PA_02095, AWI_PA_02096, AWI_PA_02097, AWI_PA_02105, AWI_PA_02106, AWI_PA_02118,
AWI_PA_02121, AWI_PA_02124, AWI_PA_02128, AWI_PA_02129, AWI_PA_02130, AWI_PA_02133, AWI_PA_02135, AWI_PA_02137,
AWI_PA_02138, AWI_PA_02140, AWI_PA_02145, AWI_PA_02146, AWI_PA_02147, AWI_PA_02148.

This publication is a contribution to the SCAR Action Group AntArchitecture, the SCAR Action Group RINGS and the Bedmap3 community project.



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
