# Peer review of "Review article: 30 years of airborne radar surveys on the Antarctic and Greenland ice sheets by the Alfred Wegener Institute"

_EGUsphere, 2025_

## Referee Comment (RC2)

Dear Editor and Authors,

Please find attached my review of Franke et al. (manuscript number: 2025-5328) with manuscript title "Review article: 30 years of airborne radar surveys on the Antarctic and Greenland ice sheets by the Alfred Wegener Institute".

This review paper and its associated data release is a great achievement for the polar radar community, and for AWI specifically who have done a lot of work to make 30 years of radar data across both the Arctic and Antarctica available to the wider scientific community. The data is fully available in open-access format, follows FAIR data principles, and is efficiently accessible through the new Radar Data over Polar Ice Sheets data viewer. Of equal importance, this review paper is very well written, the figures and tables are clear, and the text forms a great resource for current and future scientists wishing to interact with the AWI data and the science that came from it.

Together, there are now three large data providers (CRESIS, BAS and now AWI) that have released most of their aerogeophysical data over Antarctica and the Arctic in freely accessible formats. The repercussions that these data releases have on polar science cannot be understated, and I look forward to seeing other institutes follow through in future years with their own data release.

I have several minor comments regarding the text, figures or tables, but as a whole, I would recommend this paper to be published in The Cryosphere with minor revision. I very much look forward to seeing the updated version soon online.

With best wishes,

Julien Bodart
* * *
**General comments**

- **Abstract:** I think it would be useful to give some general statistics of data coverage for the Arctic and Antarctica in the abstract. For example, line km, or area covered, etc (see Line 320 for similar statistics, although these could also be provided in % or other units so that it is easy to assess the footprint of the AWI data compared with other institutes across Antarctica). This will help the reader appreciate the importance of the accompanying data release.

- **Wider geophysical data:** I find it interesting that the authors mention seldomly gravity and magnetics, or onboard LIDAR (briefly mentioned in Lines 335-336 or Section 5.7, or the Conclusion). Could the authors expand here on why the paper (and the accompanying data release) focus mainly on radar data and not on the full suite of geophysical instruments that were potentially onboard during radar data acquisition? This is of interest to the wider geophysical community and a few words at the start of the paper as to why the authors decided to focus mainly on radar would maybe be useful. Along similar lines, is there plans to release this wider geophysical data (i.e., magnetics or gravity) in the future? Section 8 mentions laser scanner and orthophotos, but no mentions of magnetics or gravity.
* * *
**Line-by-line comments:**

- Line 8: "To support scientific progress, [...]" – it would maybe be more impactful to say, "As part of this paper, and to support scientific progress, [...]" – this also reflects the point made on Lines 40-41 of the Introduction.
- Line 10: add "data" between "FAIR" and "principles". Also perhaps worth stating what FAIR stands for in the abstract. You could also reference the Wilkinson et al. 2016 paper (https://doi.org/10.1038/sdata.2016.18) for the FAIR data principles.
- Lines 24-25: Group the references by theme (e.g.: "ice thickness (Robin et al., 1969, Steinhage et al., 2001)", ice stratigraphy (Bingham et al., 2015; Winter et al., 2019; Bodart et al., 2021), etc.). Right now, it looks like the references are attached to "COF".
- Line 33: "ice sheet**s**"
- Line 48: "**p**olar aircraft**s**"
- Line 94: add again "respectively" to specify which resolution applies to which pulse mode.
- Section 3: is there a specific logic to the ordering of the radar systems in Table 1 and in the sub-sections under Section 3? Would it not make more sense to group these sub-sections by radar type (e.g., EMR, followed by UWB MCoRDS), and then group the snow and accu radars next to each other? I can't see in the text a clearly defined reason that is given for sorting the radar systems in this way, aside from the fact that this is how they are grouped in Table 1. One could argue that they could be ordered by type (as I mentioned here), or perhaps a new sub-section ordering system, consisting of "shallow" vs "deep ice" radars could be used, though admittedly this might complicate things as some radars send a shallow and deeper pulse. I note that this suggestion seems to be similar to the way that Figures 4 (snow radars), 5 (shallow radars) and 6 (deep radars) are organised, so perhaps there's room for improving the order of the sections here. I leave it up to the authors, but perhaps also making this ordering system clear in the first paragraph of Section 3 would help guide the reader, and sticking with it throughout when mentioning the radar systems. Additionally, I'd modify Figure 8 which doesn't seem to match the ordering of the radar systems from Section 3 or Table 1.
- Table 1 caption: CRESIS abbreviation is already mentioned in caption of Figure 1.
- Line 175: FAIR is still not described in the paper until this point. I'd suggest stating what the abbreviation stands for in both the abstract and here where it is first used.
- Line 183: "where other radar systems of AWI were also used" – it would be useful to show a table of where and when multiple systems were used together. For example, EMR + ASIRAS as you mention in Line 183. I'm thinking of a simplified version of Table 1 of Frémand, Bodart et al., 2023; ESSD. This is somewhat similar to Figure 8 or Table A2 (which are really useful), but it would help the reader know quickly and efficiently the specific field seasons in which multiple systems were used together. For example, on Line 339, you mention that gravity instruments were used alongside the EMR system since 2022 and then UWB+gravity later. A short table describing this, which would complement Figure 8, would in my opinion be useful in the paper.
- Lines 203-205: "The position and orientation of the aircraft is determined by four NovAtel DL-V3 GPS receivers, which sample at 20 Hz. The GPS system operates with dual-frequency tracking so that the position accuracy can be enhanced during post-processing." – is this the best place to have this information? Wouldn't it be more useful to place this in Section 2 where you describe the aircrafts (you could for example highlight the different types of GPS systems used for data georeferencing throughout the last 30 years)? Otherwise, you might need to describe the specific type of GPS that is used to georeferenced the geophysical data

for each system under Section 3 (i.e., this point can be made for Line 270 where you mention "on-board" GPS without specifying the type of GPS used).

- Line 222: You could reference Arenas-Pingarrón et al., 2023 (https://doi.org/10.1049/rsn2.12428).
- Line 253: Reference Matsuoka et al. 2025 pre-print (in review at Reviews of Geophysics; https://doi.org/10.22541/essoar.175241971.19851046/v1) instead.
- Line 285: fix "(?)"
- Line 302: "layer picking" – maybe replace layer with reflector, as it could mean the surface or bed too, presumably? If you insist on "layer" to mean isochrones, then maybe be more specific and add "englacial" or go straight for "IRHs"
- Figure 7: "Geo-Software": could you be more specific? Is this accessible to the wider public (i.e., SEGY data)? This is not mentioned in the text as far as I can see but it would be good to know that this data is available to users (this is at least what I understand from this Figure)
- Sub-section 4.2: what about the bed reflector? It's definitely worth mentioning here.
- Figure 8-9 (and anywhere else): Again, I wouldn't repeat the abbreviations again (i.e., EMR). I like 8h and 9g, this is really useful information.
- Line 329: make "In Greenland, [...]" a separate paragraph to improve readability.
- Line 336: "which, however," – a bit awkward grammar.
- Line 336-346: could the authors expand on why EMR is still in use today? I have been thinking about this for a while now, and find the reason given here (i.e., "for glaciological objectives and projects.") rather vague. This is for personal interest, but it might be of interest of the wider non-AWI community to know why EMR is still in use today and what guides AWI to still maintain it if, as the authors suggest, the "improved configuration and processing capabilities of the UWB system provide better range and along-track resolution of the bed topography"
- Lines 359:368: would this be more appropriate in Section 5.6?
- Line 375: I like this introductory sentence, and feel this is currently missing in Section 5.1 (which is historically why we have been acquiring radar data in Antarctica and Greenland)
- Line 377: add reference to Bingham et al., 2025 here too.
- Line 384: "which had a final length of 2774.15m (Wilhelms et al., 2014)." – this is a bit out of the blue information – I'd recommend removing it
- Section 5.3: you could also include Karlsson et al. (2018) here as its main focus was on using the airborne radar data to assess oldest ice. You could also add Young et al. 2017 (https://doi.org/10.5194/tc-11-1897-2017) and to another extent either in this section or another Van Liefferinge et al. (2018; https://doi.org/10.5194/tc-12-2773-2018), who use AWI data. It might be worth doing one more pass through the reference list to see if you haven't forgotten any paper that use AWI data as a significant data source in their findings.
- Line 398: "global bipolar" after "typically" implies to me like it's the case for most IRHs. I would rephrase this sentence as many IRHs are also caused by local (or at least non-bipolar/global) sources.
- Lines 449-450: Should these two examples be moved to Section 5.8: Ice shelf bathymetry and stability?
- Figure 11: you could add the new coverage from Bodart et al., 2026 (TC, in review, https://doi.org/10.5194/egusphere-2025-5381) to Figure 11c over Dronning Maud Land.
- Figure 12 caption and Lines 466-467: is it worth pointing the reader to papers which discuss these phenomenon in radar data (I'm thinking of Culberg et al., 2021 https://doi.org/10.1038/s41467-021-22656-5, for example)

- Line 519 and 531-534: I think these sentences need referencing, even if it seems like they come from the references in the sentence preceding them.
- Line 548: add "estimates" after SMB
- Line 575: You can also add the in-review paper from Bodart et al. 2026 (link above)
- Line 585: replace "up to the entire ice sheet" to "continental-scale" or similar
- Line 593: You could reference Bodart et al. 2026 here too
- Line 601: "how reliable conclusions are only possible" – improve grammar here
- Line 664: add "directly from our data portal" after "[...] for download [...]" to specify that the data still is downloadable, but from BAS and CRESIS portals instead.
- Table A2: At times in the paper, AWI coverage in the northern hemisphere is either called "Greenland" or Arctic. In the caption of Table A2, you mention "Arctic (Greenland and Canada)". It might be worth finding a way early in the manuscript to refer to this region and stick with it throughout.
- Section 7.2 (or 7.1.2): I think it might be useful to add a Table (perhaps in the SI) that lists the variables present in the netCFD files for the radar data (see Table 4 of Frémand, Bodart et al., 2023; ESSD as an example). This would be really useful for users who might not be familiar with the structure of AWI radar data (e.g., you often include the AGC data product in the files, without really explaining what this is except for a brief mention in Section 7.1.2). I know the netCDF files have metadata, but still, I think this has its place somewhere in the paper particularly for future users of your data.